# THE DIFFERENTIABLE CROSS-ENTROPY METHOD

## ABSTRACT

We study the Cross-Entropy Method (CEM) for the non-convex optimization of a continuous and parameterized objective function and introduce a differentiable variant (DCEM) that enables us to differentiate the output of CEM with respect to the objective function's parameters. In the machine learning setting this brings CEM inside of the end-to-end learning pipeline where this has otherwise been impossible. We show applications in a synthetic energy-based structured prediction task and in non-convex continuous control. In the control setting we show on the simulated cheetah and walker tasks that we can embed their optimal action sequences with DCEM and then use policy optimization to fine-tune components of the controller as a step towards combining model-based and model-free RL.

## 1 INTRODUCTION

Recent work in the machine learning community has shown how optimization procedures can create new building-blocks for the end-to-end machine learning pipeline (Gould et al., 2016; Johnson et al., 2016; Amos et al., 2017; Amos & Kolter, 2017; Domke, 2012; Metz et al., 2016; Finn et al., 2017; Belanger et al., 2017; Rusu et al., 2018; Srinivas et al., 2018; Amos et al., 2018). In this paper we focus on the setting of optimizing an *unconstrained*, *non-convex*, and *continuous* objective function $f_\theta(x) : \mathbb{R}^n \times \Theta \to \mathbb{R}$ as $\hat{x} = \arg\min_x f_\theta(x)$, where $f$ is parameterized by $\theta \in \Theta$ and has inputs $x \in \mathbb{R}^n$. If it exists, *some* (sub-)derivative $\nabla_\theta \hat{x}$ is useful in the machine learning setting to make the output of the optimization procedure end-to-end learnable. For example, $\theta$ could parameterize a predictive model that is generating potential outcomes conditional on $x$ happening that you want to optimize over. End-to-end learning in these settings can be done by defining a loss function $\mathcal{L}$ on top of $\hat{x}$ and taking gradient steps $\nabla_\theta \mathcal{L}$. If $f_\theta$ were *convex* this gradient is easy to analyze and compute when it exists and is unique (Gould et al., 2016; Johnson et al., 2016; Amos et al., 2017; Amos & Kolter, 2017). Unfortunately analyzing and computing a "derivative" through the *non-convex* $\arg\min$ here is not as easy and is challenging in theory and practice. No such derivative may exist in theory, it might not be unique, and even if it uniquely exists, the numerical solver being used to compute the solution may not find a global or even local optimum of $f$. One promising direction to sidestep these issues is to approximate the $\arg\min$ operation with an explicit optimization procedure that is interpreted as just another compute graph and unrolled through. This is most commonly done with gradient descent as in Domke (2012); Metz et al. (2016); Finn et al. (2017); Belanger et al. (2017); Rusu et al. (2018); Srinivas et al. (2018); Foerster et al. (2018). This approximation adds significant definition and structure to an otherwise extremely ill-defined desiderata at the cost of biasing the gradients and enabling the learning procedure to over-fit to the hyper-parameters of the optimization algorithm, such as the number of gradient steps or the learning rate.

In this paper we show that the *Cross-Entropy Method* (CEM) (De Boer et al., 2005) is a reasonable alternative to unrolling gradient descent for approximating the derivative through an *unconstrained*, *non-convex*, and *continuous* $\arg\min$. CEM for optimization is a *zeroth-order optimizer* and works by generating a sequence of samples from the objective function. We show a simple and computationally negligible way of making CEM differentiable that we call DCEM by using the smooth top-$k$ operation from Amos et al. (2019). This also brings CEM into the end-to-end learning process in cases where there is otherwise a disconnection between the objective that is being learned and the objective that is induced by deploying CEM on top of those models.

We first quickly study DCEM in a simple non-convex *energy-based learning* setting for regression. We contrast using unrolled gradient descent and DCEM for optimizing over a SPEN (Belanger & McCallum, 2016). We show that unrolling through gradient descent in this setting over-fits to the number of gradient steps taken and that DCEM generates a more reasonable energy surface.

Our main application focus is on using DCEM in the context of *non-convex continuous control*. This setting is especially interesting as vanilla CEM is the state-of-the-art method for solving the control optimization problem with *neural network transition dynamics* as in Chua et al. (2018); Hafner et al. (2018). We show that DCEM is useful for embedding *action sequences* into a lower-dimensional space to make solving the control optimization process significantly less computationally and memory expensive. This gives us a controller that induces a *differentiable policy class* parameterized by the model-based components. We then use PPO (Schulman et al., 2017) to *fine-tune* the model-based components, demonstrating that it *is* possible to use standard policy learning for model-based RL in addition to just doing maximum-likelihood fitting to observed trajectories.

## 2 BACKGROUND AND RELATED WORK

### 2.1 DIFFERENTIABLE OPTIMIZATION-BASED MODELING IN MACHINE LEARNING

*Optimization-based modeling* is a way of integrating specialized operations and domain knowledge into end-to-end machine learning pipelines, typically in the form of a parameterized $\arg\min$ operation. *Convex*, *constrained*, and *continuous* optimization problems, *e.g.* as in Gould et al. (2016); Johnson et al. (2016); Amos et al. (2017); Amos & Kolter (2017), capture many standard layers as special cases and can be differentiated through by applying the *implicit function theorem* to a set of optimality conditions from convex optimization theory, such as the *KKT conditions*. *Non-convex* and *continuous* optimization problems, *e.g.* as in Domke (2012); Belanger & McCallum (2016); Metz et al. (2016); Finn et al. (2017); Belanger et al. (2017); Rusu et al. (2018); Srinivas et al. (2018); Foerster et al. (2018); Amos et al. (2018); Pedregosa (2016); Jenni & Favaro (2018); Rajeswaran et al. (2019), are more difficult to differentiate through. Differentiation is typically done by *unrolling gradient descent* or applying the implicit function theorem to *some* set of optimality conditions, sometimes forming a locally convex approximation to the larger non-convex problem. *Unrolling gradient descent* is the most common way and approximates the $\arg\min$ operation with gradient descent for the forward pass and interprets the operations as just another compute graph for the backward pass that can all be differentiated through. In contrast to these works, we show how *continuous* and *nonconvex* $\arg\min$ operations can also be approximated with the *cross entropy method* (De Boer et al., 2005) as an alternative to unrolling gradient descent.

### 2.2 EMBEDDING DOMAINS FOR OPTIMIZATION PROBLEMS

Oftentimes the *solution space* of high-dimensional optimization problems may have structural properties that an optimizer can exploit to find a better solution or to find the solution quicker than an otherwise naïve optimizer. This is done in the context of meta-learning in Rusu et al. (2018) where gradient-descent is unrolled over a latent space. In the context of Bayesian optimization this has been explored with random feature embeddings, hand-coded embeddings, and auto-encoder-learned embeddings (Antonova et al., 2019; Oh et al., 2018; Calandra et al., 2016; Wang et al., 2016; Garnett et al., 2013; Ben Salem et al., 2019; Kirschner et al., 2019). We show that DCEM is another reasonable way of learning an embedded domain for exploiting the structure in and efficiently solving larger optimization problems, with the significant advantage of DCEM being that the latent space is directly learned to be optimized over as part of the end-to-end learning pipeline.

### 2.3 RL AND CONTROL

High-dimensional non-convex optimization problems that have a lot of structure in the solution space naturally arise in the control setting where the controller seeks to optimize the same objective in the same controller dynamical system from different starting states. This has been investigated in, *e.g.*, planning (Ichter et al., 2018; Ichter & Pavone, 2019; Mukadam et al., 2018; Kurutach et al., 2018; Srinivas et al., 2018; Yu et al., 2019; Lynch et al., 2019), and policy distillation (Wang & Ba, 2019). Chandak et al. (2019) shows how to learn an action space for model-free learning and

Co-Reyes et al. (2018); Antonova et al. (2019) embed *action sequences* with a VAE. There has also been a lot of work on learning reasonable latent *state space* representations (Tasfi & Capretz, 2018; Zhang et al., 2018; Gelada et al., 2019; Miladinović et al., 2019) that may have structure imposed to make it more controllable (Watter et al., 2015; Banijamali et al., 2017; Ghosh et al., 2018; Anand et al., 2019; Levine et al., 2019; Singh et al., 2019). In contrast to these works, we learn how to encode action sequences directly with DCEM instead of auto-encoding the sequences. This has the advantages of 1) never requiring the expensive expert's solution to the control optimization problem, 2) potentially being able to surpass the performance of an expert controller that uses the full action space, and 3) being end-to-end learnable through the controller for the purpose of finding a latent space of sequences that DCEM is good at searching over.

Another direction the RL and control has been pursuing is on the combination of model-based and model-free methods (Bansal et al., 2017; Okada et al., 2017; Jonschkowski et al., 2018; Pereira et al., 2018; Amos et al., 2018; Okada & Taniguchi, 2019; Janner et al., 2019; Pong et al., 2018). Amos et al. (2018) proposes differentiable MPC and only do imitation learning on the cartpole and pendulum tasks with known or lightly-parameterized dynamics — in contrast, we are able to 1) scale our differentiable controller up to the cheetah and walker tasks, 2) use neural network dynamics inside of our controller, and 3) backpropagate a policy loss through the output of our controller and into the internal components.

## 3    THE DIFFERENTIABLE CROSS-ENTROPY METHOD (DCEM)

We focus on uses of the Cross-Entropy Method (CEM) (De Boer et al., 2005) for optimization in this paper. In this setting, suppose we have a *non-convex*, *deterministic*, and *continuous* objective function $f_\theta(x)$ *parameterized* by $\theta$ over a domain $\mathbb{R}^n$ and we want to solve the optimization problem

$$\hat{x} = \arg\min_x f_\theta(x) \tag{1}$$

The original form of CEM is an *iterative* and *zeroth-order* algorithm to approximate the solution of eq. (1) with a sequence of samples from a sequence of *parametric sampling distributions* $g_\phi$ defined over the domain $\mathbb{R}^n$, such as Gaussians.

We refer the reader to De Boer et al. (2005) for more details and motivations for using CEM and briefly describe how it works here. Given a *sampling distribution* $g_\phi$, the hyper-parameters of CEM are the number of *candidate points* sampled in each iteration $N$, the number of *elite candidates* $k$ to use to fit the new sampling distribution to, and the number of iterations $T$. The iterates of CEM are the *parameters* $\phi$ of the sampling distribution. CEM starts with an *initial* sampling distribution $g_{\phi_1}(X) \in \mathbb{R}^n$, and in each iteration $t$ generates $N$ samples from the domain $[X_{t,i}]_{i=1}^N \sim g_{\phi_t}(\cdot)$, evaluates the function at those points $v_{t,i} = f_\theta(X_{t,i})$, and re-fits the sampling distribution to the top-$k$ samples by solving the maximum-likelihood problem[1]

$$\phi_{t+1} = \arg\max_\phi \sum_i \mathbb{1}\{v_{t,i} \leq \pi(v_t)_k\} \log g_\phi(X_{t,i}), \tag{2}$$

where the indicator $\mathbb{1}\{P\}$ is 1 if $P$ is true and 0 otherwise, $g_\phi(X)$ is the likelihood of $X$ under $g_\theta$ and $\pi(x)$ sorts $x \in \mathbb{R}^n$ in ascending order so that $\pi(x)_1 \leq \pi(x)_2 \leq \ldots \leq \pi(x)_n$. We can then map from the final distribution $g_{\phi_T}$ back to the domain $\mathbb{R}^n$ by taking the mean of it, *i.e.* $\hat{x} = \mathbb{E}[g_{\phi_{T+1}}(\cdot)]$.

**Proposition 1.** *For multivariate isotropic Gaussian sampling distributions we have that $\phi = \{\mu, \sigma^2\}$ and eq. (2) has a closed-form solution given by the sample mean and variance of the top-$k$ samples as $\mu_{t+1} = 1/k \sum_{i \in \mathcal{I}_t} X_{t,i}$ and $\sigma_{t+1}^2 = 1/k \sum_{i \in \mathcal{I}_t} (X_{t,i} - \mu_{t+1})^2$, where the top-$k$ indexing set is $\mathcal{I}_t = \{i : v_{t,i} \leq \pi(v_t)_k\}$.*

This is well-known in statistics and is discussed in, *e.g.*, Friedman et al. (2001).

---

[1]The Cross-Entropy Method's name comes from eq. (2) more generally optimizing the cross-entropy measure between two distributions.

---

**Algorithm 1** DCEM($f_\theta, g_\phi, \phi_1; \tau, N, k, T$)

DCEM minimizes a parameterized objective function $f_\theta$ and is differentiable w.r.t. $\theta$. Each DCEM iteration samples from the distribution $g_\phi$, starting with $\phi_1$.

---

**for** t = 1 to T **do**
$\quad [X_{t,i}]_{i=1}^N \sim g_{\phi_t}(\cdot)$ $\qquad\qquad\qquad\qquad\qquad\qquad$ ▷ Sample $N$ points from the domain
$\quad v_{t,i} = f_\theta(X_{t,i})$ $\qquad\qquad\qquad\qquad\qquad$ ▷ Evaluate the objective function at those points
$\quad \mathcal{I}_t = \Pi_{\mathcal{L}_k}(v_t/\tau)$ $\qquad\qquad\qquad$ ▷ Compute the soft top-$k$ projection of the values with eq. (4)
$\quad$ Update $\phi_{t+1}$ by solving the maximum weighted likelihood problem in eq. (5)
**end for**
**return** $\mathbb{E}[g_{\phi_{T+1}}(\cdot)]$

---

Differentiating through CEM's output with respect to the objective function's parameters with $\nabla_\theta \hat{x}$ is useful, *e.g.*, to bring CEM into the end-to-end learning process in cases where there is otherwise a disconnection between the objective that is being learned and the objective that is induced by deploying CEM on top of those models. Unfortunately in the vanilla form presented above the top-$k$ operation in eq. (2) makes $\hat{x}$ non-differentiable with respect to $\theta$. The function samples can usually be differentiated through with some estimator (Mohamed et al., 2019) such as the *reparameterization trick* (Kingma & Welling, 2013), which we use in all of our experiments.

The top-$k$ operation can be made differentiable by replacing it with a soft version as done in Martins & Kreutzer (2017); Malaviya et al. (2018); Amos et al. (2019), or by using a stochastic oracle as in **?**. Here we use the *Limited Multi-Label Projection (LML) layer* (Amos et al., 2019), which projects points from $\mathbb{R}^n$ onto the *LML polytope* defined by

$$\mathcal{L}_{n,k} = \{p \in \mathbb{R}^n \mid 0 \leq p \leq 1 \text{ and } \mathbf{1}^\top p = k\}, \tag{3}$$

which is the set of points in the unit $n$-hypercube with coordinates that sum to $k$. Notationally, if $n$ is implied by the context we will leave it out and write $\mathcal{L}_k$. We propose a *temperature-scaled* LML variant to project onto the interior of the LML polytope with

$$\Pi_{\mathcal{L}_k}(x/\tau) = \underset{0<y<1}{\arg\min} \quad -x^\top y - \tau H_b(y) \quad \text{s.t.} \quad \mathbf{1}^\top y = k \tag{4}$$

where $\tau > 0$ is the temperature parameter and $H_b(y) = -\sum_i y_i \log y_i + (1 - y_i) \log(1 - y_i)$ is the binary entropy function. Equation (4) is a convex optimization layer and can be solved in a negligible amount of time with a GPU-amenable bracketing method on the univariate dual and quickly backpropagated through with implicit differentiation. We can use the LML layer to make a soft and differentiable version of eq. (2) as

$$\phi_{t+1} = \underset{\phi}{\arg\max} \sum_i \mathcal{I}_{t,i} \log g_\phi(X_{t,i}) \quad \text{subject to} \quad \mathcal{I}_t = \Pi_{\mathcal{L}_k}(v_t/\tau). \tag{5}$$

This is now a *maximum weighted likelihood* estimation problem (Markatou et al., 1997; 1998; Wang, 2001; Hu & Zidek, 2002), which still admits an analytic closed-form solution in many cases, *e.g.* for the natural exponential family (De Boer et al., 2005). Thus using the soft top-$k$ operation with the reparameterization trick, *e.g.*, on the samples from $g$ results in a differentiable variant of CEM that we call DCEM and summarize in alg. 1. We note that we usually also normalize the values in each iteration to help separate the scaling of the values from the temperature parameter.

**Proposition 2.** *The temperature-scaled LML layer $\Pi_{\mathcal{L}_k}(x/\tau)$ approaches the hard top-$k$ operation as $\tau \to 0^+$ when all components of $x$ are unique.*

We prove this in app. A by using the KKT conditions of eq. (4).

**Corollary 1.** *DCEM captures CEM as a special case as $\tau \to 0^+$.*

**Proposition 3.** *With an isotropic Gaussian sampling distribution, the update in eq. (5) becomes $\mu_{t+1} = 1/k \sum_i \mathcal{I}_{t,i} X_{t,i}$ and $\sigma_{t+1}^2 = 1/k \sum_i \mathcal{I}_{t,i} \left(X_{t,i} - \mu_{t+1}\right)^2$, where the soft top-$k$ indexing set is $\mathcal{I}_t = \Pi_{\mathcal{L}_k}(v_t/\tau)$.*

This can be proved by differentiating eq. (5), as discussed in, *e.g.*, (De Boer et al., 2005). As a corollary, this captures prop. 1 as $\tau \to 0^+$.

## 4 APPLICATIONS

### 4.1 ENERGY-BASED LEARNING

*Energy-based learning* for regression and classification estimate the conditional probability $\mathbb{P}(y|x)$ of an output $y \in \mathcal{Y}$ given an input $x \in \mathcal{X}$ with a parameterized energy function $E_\theta(y|x) \in \mathcal{Y} \times \mathcal{X} \to \mathbb{R}$ such that $\mathbb{P}(y|x) \propto \exp\{-E_\theta(y|x)\}$. *Predictions* are made by solving the optimization problem

$$\hat{y} = \arg\min_y E_\theta(y|x). \tag{6}$$

Historically linear energy functions have been well-studied, *e.g.* in Taskar et al. (2005); LeCun et al. (2006), as it makes eq. (6) easier to solve and analyze. More recently non-convex energy functions that are parameterized by neural networks are being explored — a popular one being *Structured Prediction Energy Networks* (SPENs) (Belanger & McCallum, 2016) which propose to model $E_\theta$ with neural networks. Belanger et al. (2017) suggests to do supervised learning of SPENs by approximating eq. (6) with gradient descent that is then unrolled for $T$ steps, *i.e.* by starting with some $y_0$, making gradient updates $y_{t+1} = y_t + \gamma \nabla_y E_\theta(y_t|x)$ resulting in an output $\hat{y} = y_T$, defining a loss function $\mathcal{L}$ on top of $\hat{y}$, and doing learning with gradient updates $\nabla_\theta \mathcal{L}$ that go through the inner gradient steps.

In this context we can alternatively use DCEM to approximate eq. (6). One potential consideration when training deep energy-based models with approximations to eq. (6) is the impact and bias that the approximation is going to have on the energy surface. We note that for gradient descent, *e.g.*, it may cause the energy surface to overfit to the number of gradient steps so that the output of the approximate inference procedure isn't even a local minimum of the energy surface. One potential advantage of DCEM is that the output is more likely to be near a local minimum of the energy surface so that, *e.g.*, more test-time iterations can be used to refine the solution. We empirically illustrate the impact of the optimizer choice on a synthetic example in sect. 5.1.

### 4.2 CONTROL AND REINFORCEMENT LEARNING

Our main application focus is in the *continuous control* setting where we show how to use DCEM to learn a *latent control space* that is easier to solve than the original problem *and* induces a *differentiable policy class* that allows parts of the controller to be fine-tuned with auxiliary policy or imitation losses.

We are interested in controlling *discrete-time* dynamical systems with *continuous* state-action spaces. Let $H$ be the *horizon length* of the controller and $\mathcal{U}^H$ be the space of control sequences over this horizon length, *e.g.* $\mathcal{U}$ could be a multi-dimensional real space or box therein and $\mathcal{U}^H$ could be the Cartesian product of those spaces representing the sequence of controls over $H$ timesteps. We are interested in repeatedly solving the control optimization problem[2]

$$\hat{u}_{1:H} = \arg\min_{u_{1:H} \in \mathcal{U}^H} \sum_{t=1}^{H} C_t(x_t, u_t) \text{ subject to } x_{t+1} = f^{\text{trans}}(x_t, u_t), \ x_1 = x_{\text{init}}, \tag{7}$$

---

[2]For notational convenience we omit some explicit variables from the $\arg\min$ operator when they can be inferred by the context and not used elsewhere.

---

**Algorithm 2** Learning an embedded control space with DCEM

---

**Fixed Inputs:** Dynamics $f^{\text{trans}}$, per-step state-action cost $C_t(x_t, u_t)$ (inducing $C_\theta(z; x_{\text{init}})$) horizon $H$, full control space $\mathcal{U}^H$, distribution over initial states $\mathcal{D}$
**Learned Inputs:** Decoder $f_\theta^{\text{dec}} : \mathcal{Z} \to \mathcal{U}^H$

---

**while** not converged **do**
    Sample initial state $x_{\text{init}} \sim \mathcal{D}$
    $\hat{z} = \arg\min_{z \in \mathcal{Z}} C_\theta(z; x_{\text{init}})$                  ▷ Solve the embedded control problem eq. (8)
    $\theta \leftarrow$ grad-update$(\nabla_\theta C_\theta(\hat{z}))$        ▷ Update the decoder to improve the controller's cost
**end while**

---

where we are in an initial system state $x_{\text{init}}$ governed by deterministic system *transition dynamics* $f^{\text{trans}}$, and wish to find the optimal sequence of actions $\hat{u}_{1:H}$ such that we find a valid trajectory $\{x_{1:H}, u_{1:H}\}$ that optimizes the cost $C_t(x_t, u_t)$. Typically these controllers are used for *receding horizon* control (Mayne & Michalska, 1990) where only the first action $u_1$ is deployed on the real system, a new state is obtained from the system, and the eq. (7) is solved again from the new initial state. In this case we can say the controller induces a *policy* $\pi(x_{\text{init}}) \equiv \hat{u}_1$[3] that solves eq. (7) and depends on the cost and transition dynamics, and potential parameters therein. In all of the cases we consider $f^{\text{trans}}$ is deterministic, but may be approximated by a stochastic model for learning. Some *model-based* reinforcement learning settings consider cases where $f^{\text{trans}}$ and $C$ are parameterized and potentially used in conjunction with another policy class.

For sufficiently complex dynamical systems, eq. (7) is computationally expensive and numerically instable to solve and rife with sub-optimal local minima. The Cross-Entropy Method is the state-of-the-art method for solving eq. (7) with neural network transitions $f^{\text{trans}}$ (Chua et al., 2018; Hafner et al., 2018). CEM in this context samples full action sequences and refines the samples towards ones that solve the control problem. Hafner et al. (2018) uses CEM with 1000 samples in each iteration for 10 iterations with a horizon length of 12. This requires $1000 \times 10 \times 12 = 120,000$ evaluations (!) of the transition dynamics to predict the control to be taken given a system state — and the transition dynamics may use a deep recurrent architecture as in Hafner et al. (2018) or an ensemble of models as in Chua et al. (2018). One comparison point here is a model-free neural network policy takes a *single* evaluation for this prediction, albeit sometimes with a larger neural network.

The first application we show of DCEM in the continuous control setting is to learn a *latent action space* $\mathcal{Z}$ with a parameterized *decoder* $f_\theta^{\text{dec}} : \mathcal{Z} \to \mathcal{U}^H$ that maps back up to the space of *optimal* action *sequences*, which we illustrate in fig. 3. For simplicity starting out, assume that the dynamics and cost functions are known (and perhaps even the ground-truth) and that the only problem is to estimate the decoder in isolation, although we will show later that these assumptions can be relaxed. The motivation for having such a latent space and decoder is that the millions of times eq. (7) is being solved for the same dynamic system with the same cost, the solution space of *optimal* action sequences $\hat{u}_{1:H} \in \mathcal{U}^H$ has an extremely large amount of *spatial* (over $\mathcal{U}$) and *temporal* (over time in $\mathcal{U}^H$) structure that is being ignored by CEM on the full space. The space of optimal action sequences only contains the knowledge of the trajectories that matter for solving the task at hand, such as different parts of an optimal gait, and not irrelevant control sequences. We argue that CEM over the full action space wastes a lot of computation considering irrelevant action sequences and show that these can be ignored by learning a latent space of more reasonable candidate solutions here that we search over instead. Given a decoder, the control optimization problem in eq. (7) can then be transformed into an optimization problem over $\mathcal{Z}$ as

$$\hat{z} = \underset{z \in \mathcal{Z}}{\arg\min} \ \ C_\theta(z; x_{\text{init}}) \equiv \sum_{t=1}^{H} C_t(x_t, u_t) \tag{8}$$
$$\text{subject to} \ \ x_{t+1} = f^{\text{trans}}(x_t, u_t), \ \ x_1 = x_{\text{init}}, \ \ u_{1:H} = f_\theta^{\text{dec}}(z),$$

which is still a challenging non-convex optimization problem that searches over a decoder's input space to find the optimal control sequence. We illustrate what this looks like in fig. 3, and note the impact of the decoder initialization in app. C.

---

[3]For notational convenience we also omit the dependency of $u_1$ on $x_{\text{init}}$ here.

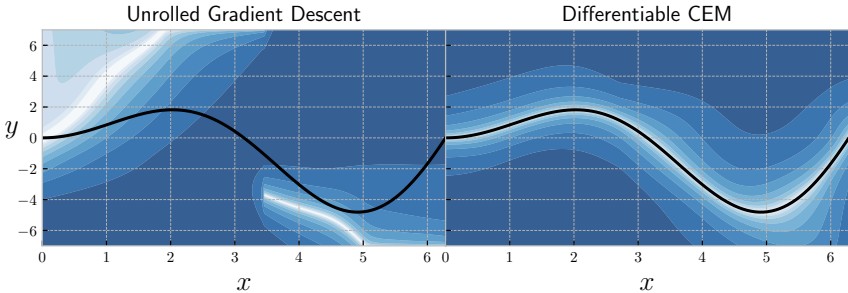

Figure 1: We trained an energy-based model with unrolled gradient descent and DCEM for a 1D regression task with the target function shown in black. Each method unrolls through 10 optimizer steps. The contour surfaces show the (normalized/log-scaled) energy surfaces, highlighting that unrolled gradient descent models can overfit to the number of gradient steps. The lighter colors show areas of lower energy.

We propose in alg. 2 to use DCEM to approximately solve eq. (8) and then *learn* the decoder directly to optimize the performance of eq. (7). Every time we solve eq. (8) with DCEM and obtain an optimal latent representation $\hat{z}$ along with the induced trajectory $\{x_t, u_t\}$, we can take a gradient step to push down the resulting cost of that trajectory with $\nabla_\theta C(\hat{z})$, which goes through the DCEM process that uses the decoder to generate samples to obtain $\hat{z}$. We note that the DCEM machinery behind this is not necessary *if* a reasonable local minima is consistently found as this is an instance of min-differentiation (Rockafellar & Wets, 2009, Theorem 10.13) but in practice this breaks down in non-convex cases when the minimum cannot be consistently found. Antonova et al. (2019); Wang & Ba (2019) solve related problems in this space and we discuss them in sect. 2.3. We also note that to learn an action embedding we still need to differentiate through the transition dynamics and cost functions to compute $\nabla_\theta C(\hat{z})$, even if the ground-truth ones are being used, since the latent space needs to have the knowledge of how the control cost will change as the decoder's parameters change.

DCEM in this setting also induces a *differentiable policy class* $\pi(x_{\text{init}}) \equiv u_1 = f^{\text{dec}}(\hat{z})_1$. This enables a policy or imitation loss $\mathcal{J}$ to be defined on the policy that can fine-tune the parts of the controller (decoder, cost, and transition dynamics) gradient information from $\nabla_\theta \mathcal{J}$. In theory the same approach could be used with CEM on the full optimization problem in eq. (7). For realistic problems without modification this is intractable and memory-intensive as it would require storing and backpropagating through every sampled trajectory, although as a future direction we note that it may be possible to delete some of the low-influence trajectories to help overcome this.

## 5 EXPERIMENTS

We use PyTorch (Paszke et al., 2017) and will openly release our DCEM library, model-based control code, and the source, plotting, and analysis code for all of our experiments.

### 5.1 UNROLLING OPTIMIZERS FOR REGRESSION AND STRUCTURED PREDICTION

In this section we briefly explore the impact of the inner optimizer on the energy surface of a SPEN as discussed in sect. 4.1. For illustrative purposes we consider a simple unidimensional regression task where the *ground-truth data* is generated from $f(x) = x\sin(x)$ for $x \in [0, 2\pi]$. We model $\mathbb{P}(y|x) \propto \exp\{-E_\theta(y|x)\}$ with a single neural network $E_\theta$ and make predictions $\hat{y}$ by solving the optimization problem eq. (6). Given the ground-truth output $y^\star$, we use the loss $\mathcal{L}(\hat{y}, y^\star) = ||\hat{y} - y^\star||_2^2$ and take gradient steps of this loss to shape the energy landscape.

We consider approximating eq. (6) with unrolled gradient descent and DCEM with Gaussian sampling distributions. Both of these are trained to take 10 optimizer steps and we use an inner learning rate of 0.1 for gradient descent and with DCEM we use 10 iterations with 100 samples per iteration and 10 elite candidates, with a temperature of 1. For both algorithms we start the initial iterate at $y_0 = 0$. We show in app. B that both of these models attain the same loss on the training dataset

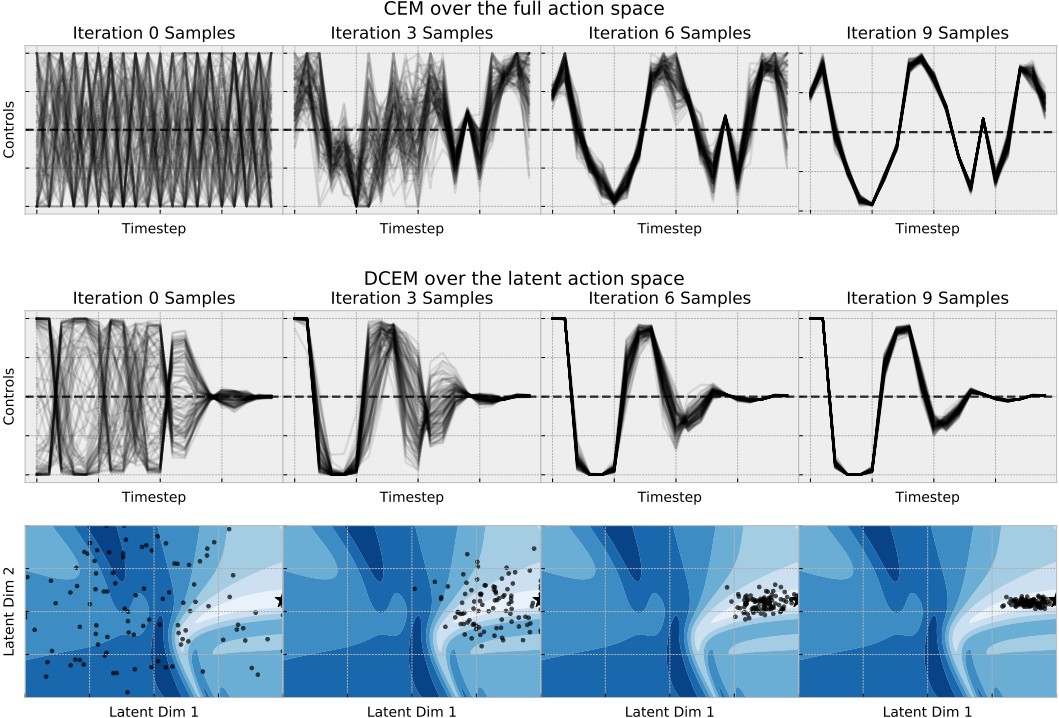

Figure 2: Visualization of the samples that CEM and DCEM generate to solve the cartpole task starting from the same initial system state. The plots starting at the top-left show that CEM initially starts with no temporal knowledge over the control space whereas embedded DCEM's latent space generates a more feasible distribution over control sequences to consider in each iteration. Embedded DCEM uses an order of magnitude less samples and is able to generate a better solution to the control problem. The contours on the bottom show the controller's cost surface $C(z)$ from eq. (8) for the initial state — the lighter colors show regions with lower costs.

but, since this is a unidimensional regression task, we can visualize the entire energy surfaces over the joint input-output space in fig. 1. This shows that gradient descent has learned to adapt from the initial $y_0 = 0$ position to the final position by descending along the function's surface as we would expect, but there is no reason why the energy surface should be a local minimum around the last iterate $\hat{y} = y_{10}$. The energy surface learned by CEM captures local minima around the regression target as the sequence of Gaussian iterates are able to capture a more global view of the function landscape and need to focus in on a minimum of it for regression. We show ablations in app. B from training for 10 inner iterations and then evaluating with a different number of iterations and show that gradient descent quickly steps away from making reasonable predictions.

**Discussion and limitations.** We note that other tricks *could* be used to force the output to be at a local minimum with gradient descent, such as using multiple starting points or randomizing the number of gradient descent steps taken — our intention here is to highlight this behavior in the vanilla case. We also note that DCEM is susceptible to overfitting to the hyper-parameters behind it in similar, albeit less obvious ways.

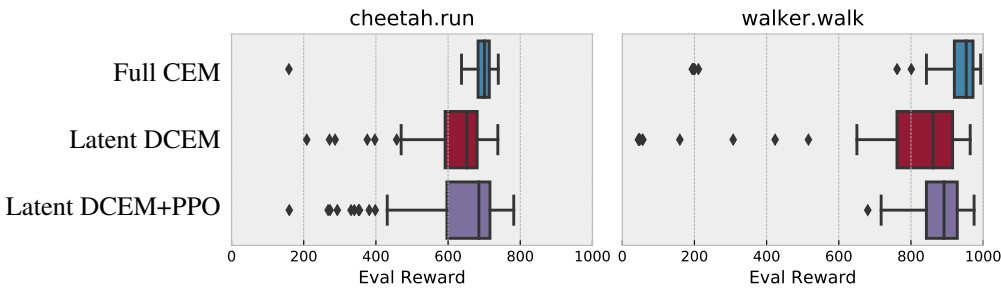

Figure 4: We evaluated our final models by running 100 episodes each on the cheetah and walker tasks. CEM over the full action space uses 10,000 trajectories for control at each time step while embedded DCEM samples only 1000 trajectories. DCEM almost recovers the performance of CEM over the full action space and PPO fine-tuning of the model-based components helps bridge the gap.

## 5.2 CONTROL

### 5.2.1 STARTING SIMPLE: EMBEDDING THE CARTPOLE'S ACTION SPACE

We first show that it is possible to learn an embedded control space as discussed in sect. 4.2 in an isolated setting. We use the standard cartpole dynamical system from Barto et al. (1983) with a *continuous* state-action space. We assume that the ground-truth dynamics and cost are *known* and use the differentiable ground-truth dynamics and cost implemented in PyTorch from Amos et al. (2018). This isolates the learning problem to *only* learning the embedding so that we can study what this is doing without the additional complications that arise from exploration, estimating the dynamics, learning a policy, and other non-stationarities. We show experiments with these assumptions relaxed in sect. 5.2.2.

We use DCEM and alg. 2 to learn a 2-dimensional latent space $\mathcal{Z} = [0, 1]^2$ that maps back up to the full control space $\mathcal{U}^H = [0, 1]^H$ where we focus on horizons of length $H = 20$. For DCEM over the embedded space we use 10 iterations with 100 samples in each iteration and 10 elite candidates, again with a temperature of 1. We show the details in app. D that we *are* able to recover the performance of an expert CEM controller that uses an order-of-magnitude more samples fig. 2 shows a visualization of what the CEM and embedded DCEM iterates look like to solve the control optimization problem from the same initial system state. CEM spends a lot of evaluations on sequences in the control space that are unlikely to be optimal, such as the ones the bifurcate between the boundaries of the control space at every timestep, while our embedded space is able to learn more reasonable proposals.

### 5.2.2 SCALING UP TO THE CHEETAH AND WALKER

Next we show that we can relax the assumptions of having known transition dynamics and reward and show that we can learn a latent control space on top of a learned model on the `cheetah.run` and `walker.walk` tasks with frame skips of 4 and 2, respectively, from the DeepMind control suite (Tassa et al., 2018) using the MuJoCo physics engine (Todorov et al., 2012). We then fine-tune the policy induced by the embedded controller with PPO (Schulman et al., 2017), sending the policy loss directly back into the reward and latent embedding modules underlying the controller.

We start with a state-of-the-art model-based RL approach by noting that the PlaNet (Hafner et al., 2018) restricted state space model (RSSM) is a reasonable architecture for proprioceptive-based control in addition to just pixel-based control. We show the graphical model we use in fig. 3, which maintains deterministic hidden states $h_t$ and stochastic (proprioceptive) system observations $x_t$ and rewards $r_t$. We model transitions as $h_{t+1} = f_\theta^{\text{trans}}(h_t, x_t)$, observations with $x_t \sim f_\theta^{\text{odec}}(h_t)$, rewards with $r_t = f_\theta^{\text{rew}}(h_t, x_t)$, and map from the latent action space to action sequences with $u_{1:T} = f^{\text{dec}}(z)$. We follow the online training procedure of Hafner et al. (2018) to initialize all of the models except for the action decoder $f^{\text{dec}}$, using approximately 2M timesteps. We then use a variant of alg. 2 to learn $f^{\text{dec}}$ to embed the action space for control with DCEM, which we also do online while updating the models. We describe the full training process in app. E.

Our DCEM controller induces a differentiable policy class $\pi_\theta(x_{\text{init}})$ where $\theta$ are the parameters of the models that impact the actions that the controller is selecting. We then use PPO to define a loss on top of this policy class and fine-tune the components (the decoder and reward module) so that they improve the episode reward rather than the maximum-likelihood solution of observed trajectories. We chose PPO because we thought it would be able to fine-tune the policy with just a few updates because the policy is starting at a reasonable point, but this did not turn out to be the case and in the future other policy optimizers can be explored. We implement this by making our DCEM controller the policy in the PyTorch PPO implementation by Kostrikov (2018). We provide more details behind our training procedure in app. E.

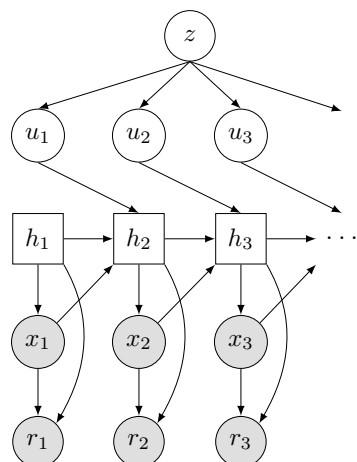

Figure 3: Our RSSM with action sequence embeddings

We evaluate our controllers on 100 test episodes and the rewards in fig. 4 show that DCEM is almost (but not exactly) able to recover the performance of doing CEM over the full action space while using an order-of-magnitude less trajectory samples (1,000 vs 10,0000). PPO fine-tuning helps bridge the gap between the performances.

Videos of our trained models are available at:

https://sites.google.com/view/diff-cross-entropy-method

**Discussion and limitations.** DCEM in the control setting has many potential future directions to explore and help bring efficiency and policy-based fine-tuning to model-based reinforcement learning. Much more analysis and experimentation is necessary to achieve this as we faced many issues getting the model-based cheetah and walker tasks to work that did not arise in the ground-truth cartpole task. We discuss this more in app. E. We also did not focus on the sample complexity of our algorithms getting these proof-of-concept experiments working. We also note that other reasonable baselines on this task could involve distilling the controller into a model-free policy and then doing search on top of that policy, as done in POPLIN (Wang & Ba, 2019).

## 6 CONCLUSIONS AND FUTURE DIRECTIONS

We have laid the foundations for differentiating through the cross-entropy method and have brought CEM into the end-to-end learning pipeline. Beyond further explorations in the energy-based learning and control contexts we showed here, DCEM can be used anywhere gradient descent is unrolled. We find this especially promising for meta-learning, potentially building on LEO (Rusu et al., 2018). Inspired by DCEM, other more powerful sampling-based optimizers could be made differentiable in the same way, potentially optimizers that leverage gradient-based information in the inner optimization steps (Sekhon & Mebane, 1998; Theodorou et al., 2010; Stulp & Sigaud, 2012; Maheswaranathan et al., 2018) or by also learning the hyper-parameters of structured optimizers (Li & Malik, 2016; Volpp et al., 2019; Chen et al., 2017).

ACKNOWLEDGMENTS

We acknowledge the scientific Python community (Van Rossum & Drake Jr, 1995; Oliphant, 2007) for developing the core set of tools that enabled this work, including PyTorch (Paszke et al., 2017), Jupyter (Kluyver et al., 2016), Matplotlib (Hunter, 2007), seaborn (Waskom et al., 2018), numpy (Oliphant, 2006; Van Der Walt et al., 2011), pandas (McKinney, 2012), and SciPy (Jones et al., 2014).

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

## A    PROOF OF PROP. 2

*Proof.* We first note that a solution exists to the projection operation, and it is unique, which comes from the strict convexity of the objective (Rao, 1984). The Lagrangian of the temperature-scaled LML projection in eq. (4) is

$$L(y, \nu) = -x^\top y - \tau H_b(y) + \nu(k - 1^\top y). \tag{9}$$

Differentiating eq. (9) gives

$$\nabla_y L(y, \nu) = -x + \tau \log \frac{y}{1-y} - \nu \tag{10}$$

and the first-order optimality condition $\nabla_y L(y^\star, \nu^\star) = 0$ gives $y_i^\star = \sigma(\tau^{-1}(x_i + \nu^*))$, where $\sigma$ is the sigmoid function. Using lem. 1 as $\tau \to 0^+$ gives

$$y_i^\star = \begin{cases} 1 & \text{if } x_i > -\nu^* \\ 0 & \text{if } x_i < -\nu^* \\ 1/2 & \text{otherwise.} \end{cases} \tag{11}$$

Substituting this back into the constraint $1^\top y^\star = k$ gives that $\pi(x)_k < -\nu^* < \pi(x)_{k+1}$, where $\pi(x)$ sorts $x \in \mathbb{R}^n$ in ascending order so that $\pi(x)_1 \leq \pi(x)_2 \leq \ldots \leq \pi(x)_n$. Thus we have that $y_i^\star = \mathbb{1}\{x_i \geq \pi(x)_k\}$, which is 1 when $x_i$ is in the top-$k$ components of $x$ and 0 otherwise, and therefore the temperature-scaled LML layer approaches the hard top-$k$ function as $\tau \to 0^+$.  □

**Lemma 1.**

$$\lim_{\tau \to 0^+} \sigma(x/\tau) = \begin{cases} 1 & \text{if } x > 0 \\ 0 & \text{if } x < 0 \\ 1/2 & \text{otherwise,} \end{cases} \tag{12}$$

*where $\sigma(x/\tau) = (1 + \exp\{-x/\tau\})^{-1}$ is the temperature-scaled sigmoid.*

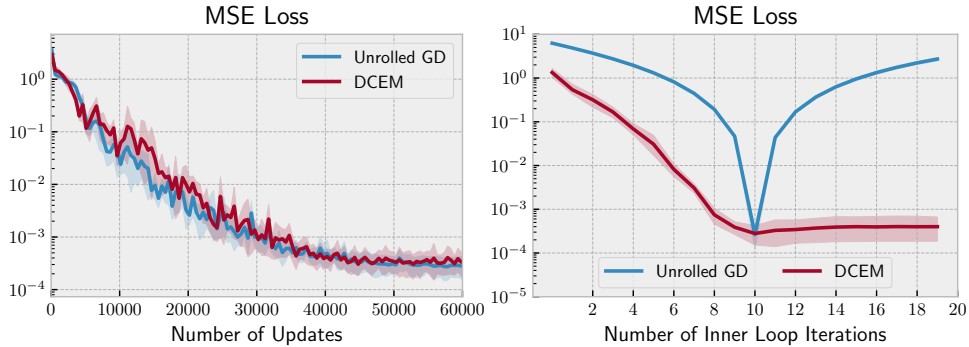

Figure 5: **Left:** Convergence of DCEM and unrolled GD on the regression task. **Right:** Ablation showing what happens when DCEM and unrolled GD are trained for 10 inner steps and then a different number of steps is used at test-time. We trained three seeds for each model and the shaded regions show the 95% confidence interval.

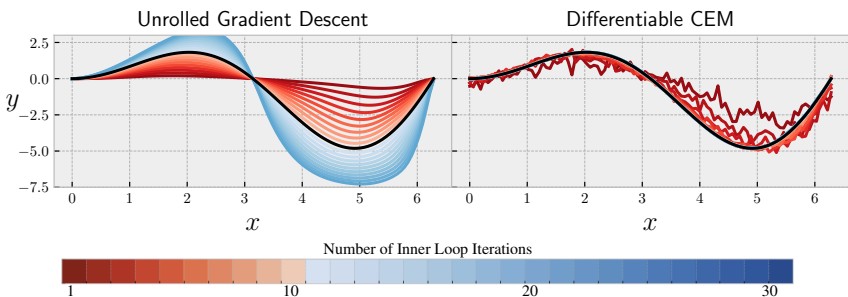

Figure 6: Visualization of the predictions made by ablating the number of inner loop iterations for unrolled GD and DCEM. The ground-truth regression target is shown in black.

## B    MORE DETAILS: SIMPLE REGRESSION TASK

Figure 5 (left) shows the convergence of unrolled GD and DCEM on the training data, showing that they are able to obtain the same training loss despite inducing very different energy surfaces. Figure 5 (right) and fig. 6 shows the impact of training gradient descent and DCEM to take 10 inner optimization steps and then ablating the number of inner steps at test-time.

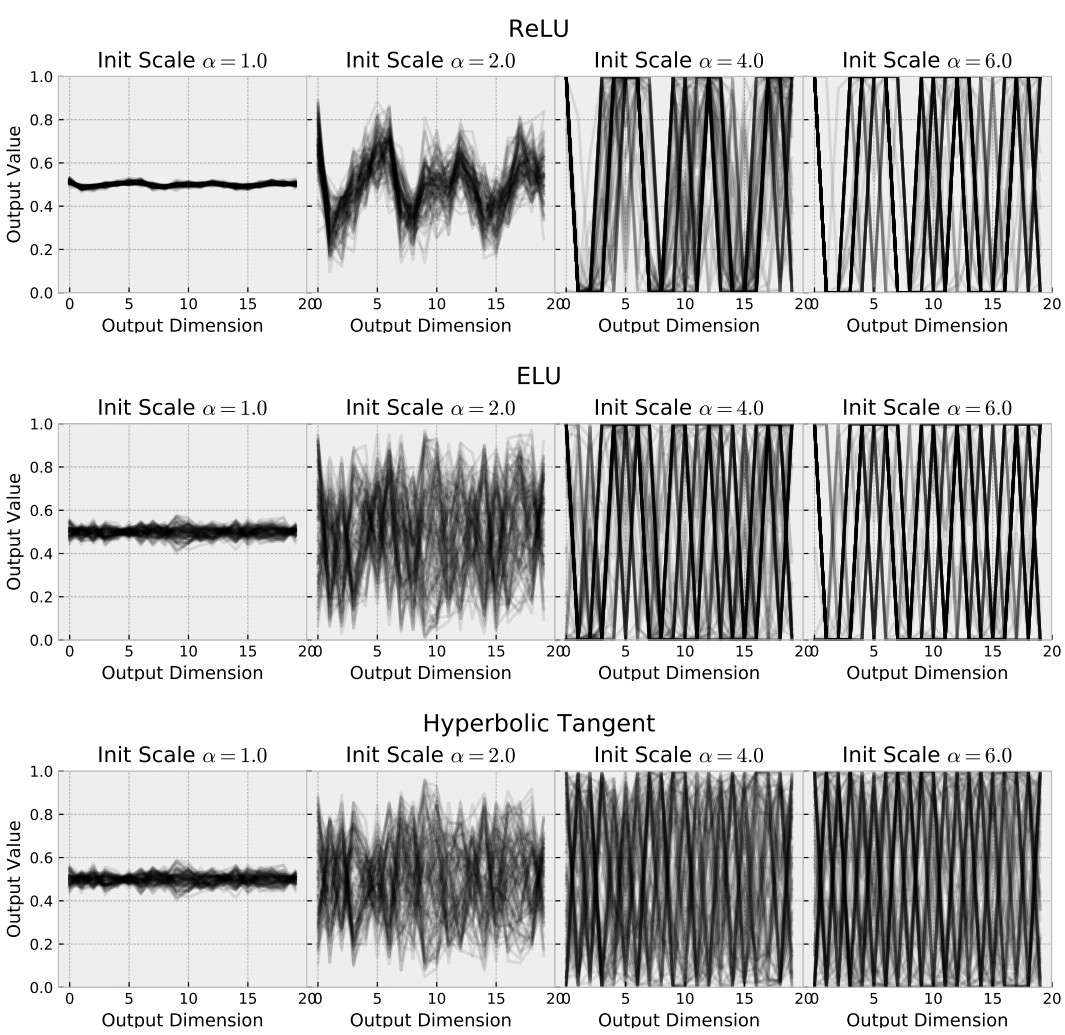

Figure 7: Impact of the activation function on the initial decoder values

## C    MORE DETAILS: DECODER INITIALIZATIONS AND ACTIVATION FUNCTIONS

We have found the decoder to be influenced by the activation function that's used with it and have found the ELU (Clevert et al., 2015) to perform the best. fig. 7 conveys some intuition behind this choice. We randomly initialize a neural network $u = f_\theta(z)$ with no biases, where $\theta = \{W_i\}_i$ for every layer weight $W_i$, and then scale the weights with $\alpha\theta$. We then sample $z \sim \mathcal{N}(0, I)$, pass them through $f_{\alpha\theta}$, and plot the outputs. The ReLU (Nair & Hinton, 2010) induces an extremely biased distribution which is seen more prevalently as $\alpha$ grows that is not as present when using the ELU or hyperbolic tangent since they are almost linear around zero. Despite the reasonable looking initializations for the hyperbolic tangent, we found that it does not perform as well in practice in our experiments. We found that the initial scale $\alpha$ of the decoder's parameters is also important for learning because of the network is not initially producing samples that cover the full output space as shown with $\alpha = 1$, it seems hard for it to learn how to expand to cover the full output space.

## D    MORE DETAILS: CARTPOLE EXPERIMENT

In this section we discuss some of the ablations we considered when learning the latent action space for the cartpole task. In all settings we use DCEM to unroll 10 inner iterations that samples 100 candidate points in each iteration and has an elite set of 10 candidates.

For training, we randomly sample initial starting points of the cartpole and for validation we use a fixed set of initial points. Figure 8 shows the convergence of models as we vary the latent space dimension and temperature parameter, and fig. 9 shows that DCEM is able to fully recover the expert performance on the cartpole. Because we are operating in the ground-truth dynamics setting we measure the performance by comparing the controller costs. We use $\tau = 0$ to indicate the case where we optimize over the latent space with vanilla CEM and then update the decoder with $\nabla_z C(f_\theta^{\mathrm{dec}}(\hat{z}))$, where the gradient doesn't go back into the optimization process that produced $\hat{z}$. This is non-convex min differentiation and is reasonable when $\hat{z}$ is near-optimal, but otherwise is susceptible to making the decoder difficult to search over.

These results show a few interesting points that come up in this setting, which of course may be different in other settings. Firstly that with a two-dimensional latent space, all of the temperature values are able to find a reasonable latent space at some point during training. However after more updates, the lower-temperature experiments start updating the decoder in ways that make it more difficult to search over and start achieving worse performance than the $\tau = 1$ case. For higher-dimensional latent spaces, the DCEM machinery is necessary to keep the decoder searchable. Furthermore we notice that just a 16-dimensional latent space for this task can be difficult for learning, one reason this could be is from DCEM having too many degrees of freedom in ways it can update the decoder to improve the performance of the optimizer.

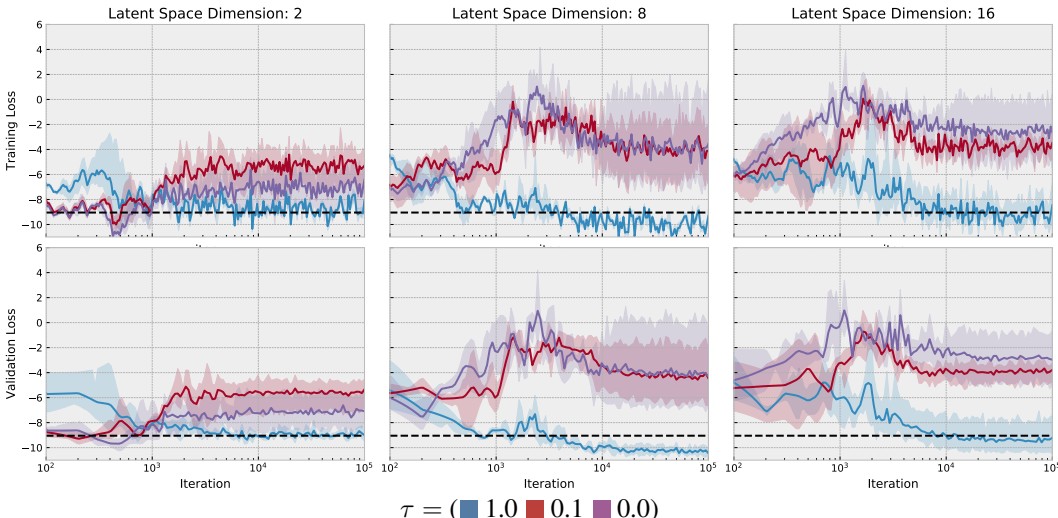

Figure 8: Training and validation loss convergence for the cartpole task. The dashed horizontal line shows the loss induced by an expert controller. Larger latent spaces seem harder to learn and as DCEM becomes less differentiable, the embedding is more difficult to learn. The shaded regions show the 95% confidence interval around three trials.

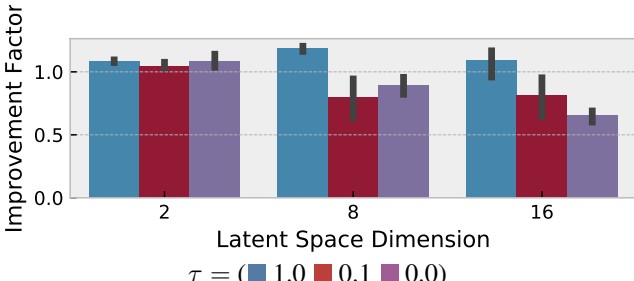

Figure 9: Improvement factor on the ground-truth cartpole task from embedding the action space with DCEM compared to running CEM on the full action space, showing that DCEM is able to recover the full performance. We use the DCEM model that achieves the best validation loss. The error lines show the 95% confidence interval around three trials.

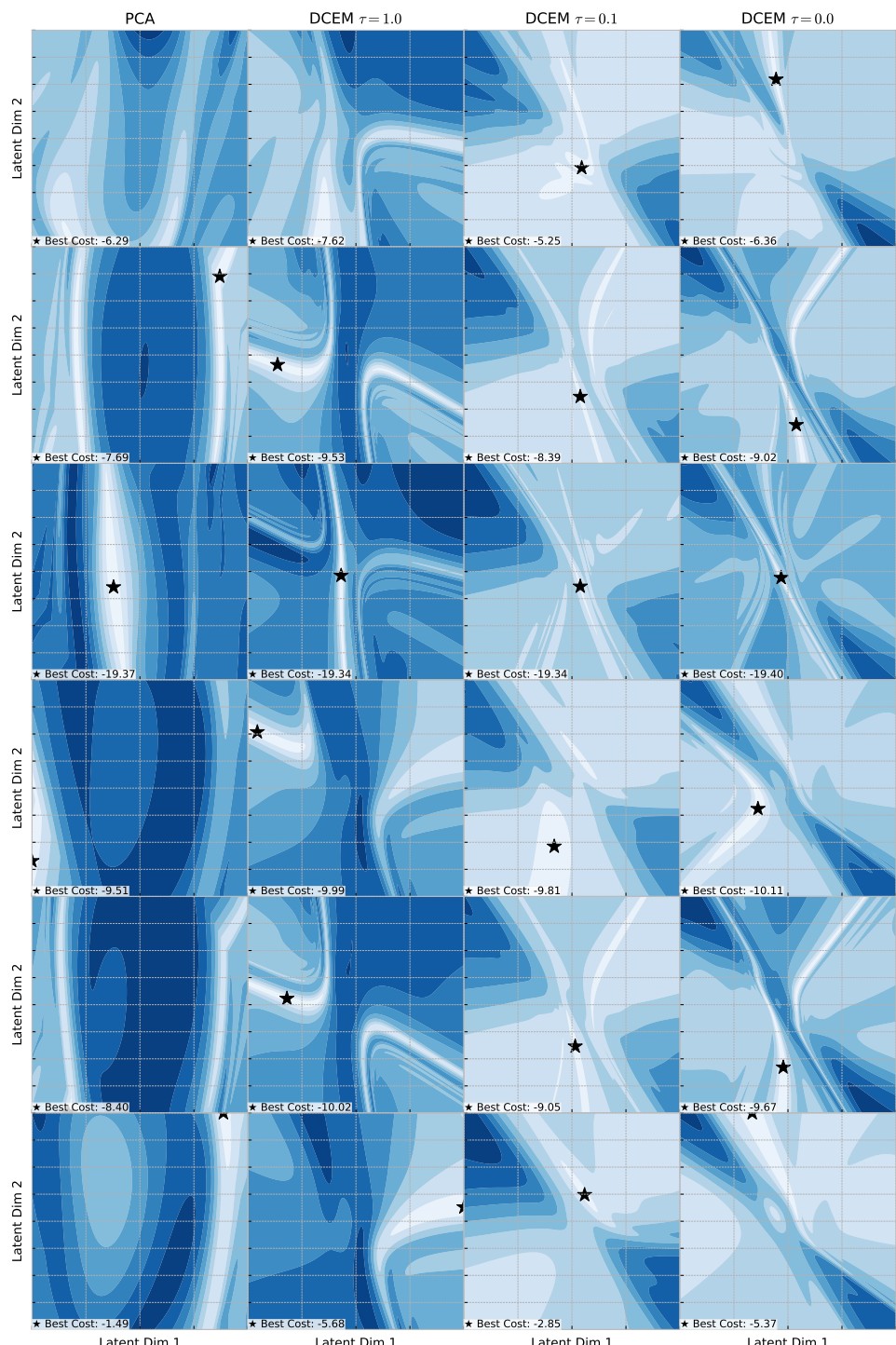

Figure 10: Learned DCEM reward surfaces for the cartpole task. Each row shows a different initial state of the system. We can see that as the temperature decreases, the latent representation can still capture near-optimal values, but they are in much narrower regions of the latent space than when $\tau = 1$.

---

**Algorithm 3** PlaNet (Hafner et al., 2018) variant that we use for proprioceptive control with optional DCEM embedding

---

    ▷ **Models:** a deterministic state model, a stochastic state model, a reward model, and (if using DCEM) an action sequence decoder.
    ▷ Initialize dataset $\mathcal{D}$ with $S$ random seed episodes.
    ▷ Initialize the transition model's deterministic hidden state $h_0$ and initialize the environment, obtaining the initial state estimate $x_0$.
    ▷ CEM-Solve can use DCEM or full CEM
    **for** $t = 1, \ldots, T$ **do**
        $u_t \leftarrow$ CEM-solve$(h_{t-1}, x_{t-1})$
        Add exploration noise $\epsilon \sim p(\epsilon)$ to the action $u_t$.
        $\{r_t, x_{t+1}, d_t\} \leftarrow$ env.step$(u_t)$                 ▷ Properly restarting if necessary
        Add $[r_t, x_t, u_t, d_t]$ to $\mathcal{D}$
        $h_t =$ update-hidden$(h_{t-1}, x_t, u_t, d_t)$
        **if** $t \equiv 0 \pmod{\text{update-interval}}$ **then**
            Sample trajectories $\tau = [r_\tau, x_\tau, u_\tau, d_\tau]_{\tau=1}^{H} \sim \mathcal{D}$ from the dataset.
            Obtain the hidden states of the $\{h_\tau, \hat{x}_\tau\}$ from the model.
            Compute the multi-step likelihood bound $\mathcal{L}(\tau, h_\tau, \hat{x}_\tau)$     ▷ (Hafner et al., 2018, eq 6.)
            $\theta \leftarrow$ grad-update$(\nabla_\theta \mathcal{L}_\theta(\tau, h_\tau, \hat{x}_\tau))$     ▷ Optimize the likelihood bound
            **if** using DCEM **then**
                $\hat{z}_\tau = \arg\min_{z \in \mathcal{Z}} C_\theta(z; h_\tau, \hat{x}_\tau)$     ▷ Solve the embedded control problem in eq. (8)
                $\theta \leftarrow$ grad-update$(\nabla_\theta \sum_\tau C_\theta(\hat{z}_\tau))$     ▷ Update the decoder
            **end if**
        **end if**
    **end for**

---

## E   MORE DETAILS: CHEETAH AND WALKER EXPERIMENTS

For the `cheetah.run` and `walker.walk` DeepMind control suite experiments we start with a modified PlaNet (Hafner et al., 2018) architecture that does not have a pixel decoder. We started with this over PETS (Chua et al., 2018) to show that this RSSM is reasonable for proprioceptive-based control and not just pixel-based control. This model is graphically shown in fig. 3 and has 1) a deterministic state model $h_t = f(h_{t-1}, x_{t-1}, u_{t-1})$, 2) a stochastic state model $x_t \sim p(x_t, h_t)$, and 3) a reward model: $r_t \sim p(r_t | h_t, x_t)$. In the proprioceptive setting, we posit that the deterministic state model is useful for multi-step training even in fully observable environments as it allows the model to "push forward" information about what is potentially going to happen in the future.

For the modeling components, we follow the recommendations in Hafner et al. (2018) and use a GRU (Cho et al., 2014) with 200 units as the deterministic path in the dynamics model and implement all other functions as two fully-connected layers, also with 200 units with ReLU activations. Distributions over the state space are isotropic Gaussians with predicted mean and standard deviation. We train the model to optimize the variational bound on the multi-step likelihood as presented in (Hafner et al., 2018) on batches of size 50 with trajectory sequences of length 50. We start with 5 seed episodes with random actions and in contrast to Hafner et al. (2018), we have found that interleaving the model updates with the environment steps instead of separating the updates slightly improves the performance, even in the pixel-based case, which we do not report results on here.

For the optimizers we either use CEM over the full control space or DCEM over the latent control space and use a horizon length of 12 and 10 iterations here. For full CEM, we sample 1000 candidates in each iteration with 100 elite candidates. For DCEM we use 100 candidates in each iteration with 10 elite candidates.

Our training procedure has the following three phases, which we set up to isolate the DCEM additions. We evaluate the models output from these training runs on 100 random episodes in fig. 4 in the main paper. Now that these ideas have been validated, promising directions of future work include trying to combine them all into a single training run and trying to reduce the sample complexity and number of timesteps needed to obtain the final model.

**Phase 1: Model initialization.** We start in both environments by launching a single training run of fig. 11 to get initial system dynamics. fig. 11 shows that these starting points converge to near-state-of-the-art performance on these tasks. These models take slightly longer to converge than in (Hafner et al., 2018), likely due to how often we update our models. We note that at this point, it would be ideal to use the policy loss to help fine-tune the components so that policy induced by CEM on top of the models can be guided, but this is not feasible to do by backpropagating through all of the CEM samples due to memory, so we instead next move on to initializing a differentiable controller that is feasible to backprop through.

**Phase 2: Embedded DCEM initialization.** Our goal in this phase is to obtain a differentiable controller that is feasible to backprop through.

Our first failed attempt to achieve this was to use offline training on the replay buffer, which would have been ideal as it would require no additional transitions to be collected from the environment. We tried using alg. 2, the same procedure we used in the ground-truth cartpole setting, to generate an embedded DCEM controller that achieves the same control cost on the replay buffer as the full CEM controller. However we found that when deploying this controller on the system, it quickly stepped off of the data manifold and failed to control it — this seemed to be from the controller finding holes in the model that causes the reward to be over-predicted.

We then used an online data collection process identical to the one we used for phase 1 to jointly learn the embedded control space while updating the models so that the embedded controller doesn't find bad regions in them. We show where the DCEM updates fit into alg. 3. One alternative that we tried to updating the decoder to optimize the control cost on the samples from the replay buffer is that the decoder can also be immediately updated after planning at every step. This seemed nice since it didn't require any additional DCEM solves, but we found that the decoder became too biased during the episode as samples at consecutive timesteps have nearly identical information.

For the hyper-parameters, we kept most of the DCEM hyper-parameters fixed throughout this phase to 100 samples, 10 elites, and a temperature $\tau = 1$. We ablated across 1) the number of DCEM iterations taken to be $\{3, 5, 10\}$, 2) deleting the replay buffer from phase 1 or not, and 3) re-initializing the model or not from phase 1. We report the best runs that we use as the starting point for the next phase in fig. 12, which achieve reasonable performance but don't match the performance of doing CEM over the full action space. These runs all use 10 DCEM iterations and both *keep* the replay buffer from phase 1. The Cheetah run keeps the models from phase 1 and the Walker re-initializes the models. The cheetah curve around timestep 600k shows, the stability here can be improved as sometimes the decoder finds especially bad regions in the model that induce extremely high losses.

**Phase 3: Policy optimization into the controller.** Finally now that we have a differentiable policy class induced by this differentiable controller we can do policy learning to fine-tune parts of it. We initially chose Proximal Policy Optimization (PPO) (Schulman et al., 2017) for this phase because we thought that it would be able to fine-tune the policy in a few iterations without requiring a good estimate of the value function, but this phase also ended up consuming many timesteps from the environment. Crucially in this phase, we do **not** do likelihood fitting at all, as our goal is to show that PPO can be used as another useful signal to update the parts of a controller — we did this to isolate the improvement from PPO but in practice we envision more unified algorithms that use both signals at the same time. Using the standard PPO hyper-parameters, we collect 10 episodes for each PPO training step and ablate across 1) the number of passes to make through these episodes $\{1, 2, 4\}$, 2) every combination of the reward, transition, and decoder being fine-tuned or frozen, 3) using a fixed variance of 0.1 around the output of the controller or learning this, 4) the learning rate of the fine-tuned model-based portions $\{10^{-4}, 10^{-5}\}$. Figure 13 shows the results of the best runs from this search.

We conclude by showing the PPO-fine-tuned DCEM iterates for solving a single control optimization problem from a random system state for the cheetah fig. 14. and walker fig. 15 tasks.

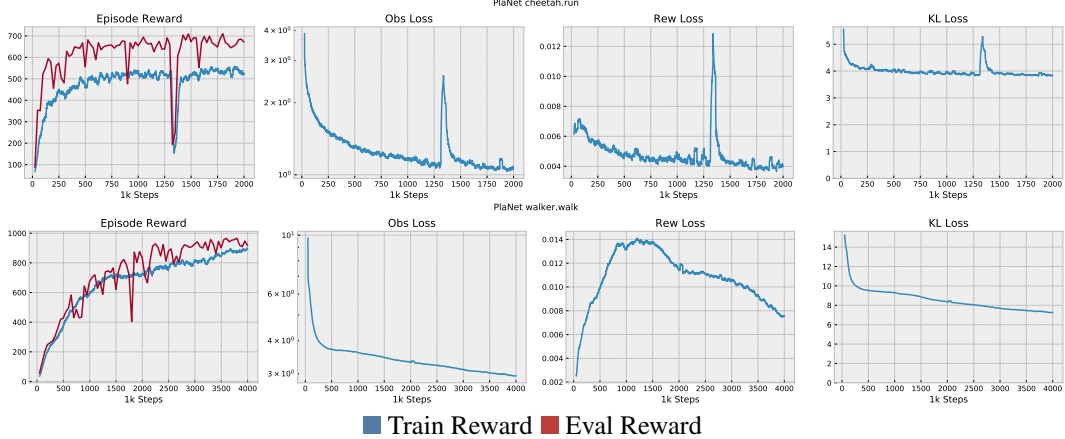

Figure 11: **Phase 1:** The two base proprioceptive PlaNet training runs that use CEM over the full action space. The evaluation loss uses 10 episodes and we show a rolling average of the training loss.

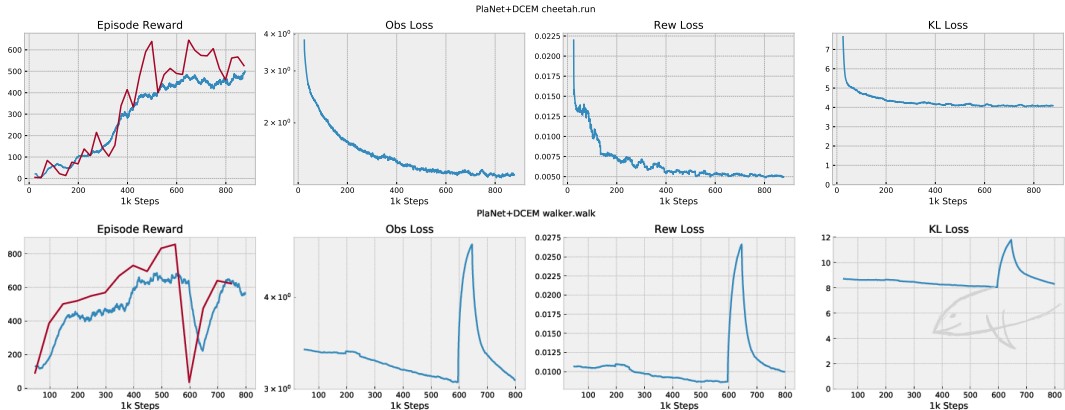

Figure 12: **Phase 2:** The training runs of learning an embedded DCEM controller with online updates. The evaluation loss uses 10 episodes and we show a rolling average of the training loss.

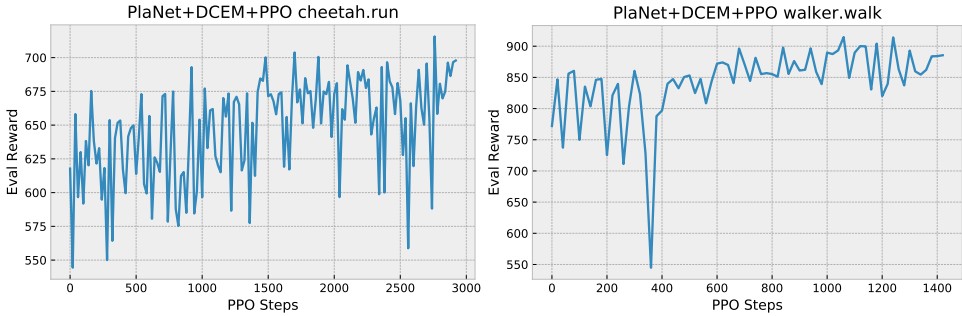

Figure 13: **Phase 3:** The training run of PPO-fine-tuning into the model-based components — we only use the PPO updates to tune these components and do optimize for the likelihood in this phase. The evaluation loss uses 10 episodes.

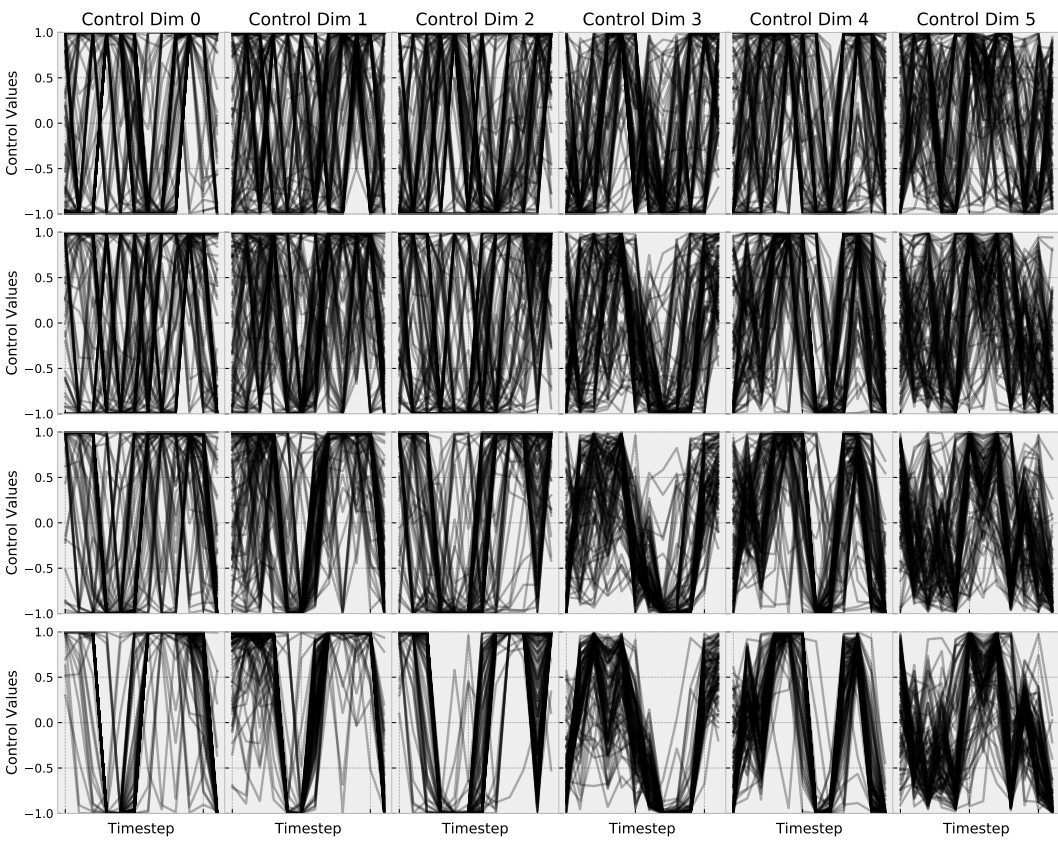

Figure 14: Visualization of the DCEM iterates on the cheetah to solve a single control problem starting from a random initial system state The rows show iterates 1, 5, 7, 10 from the top to bottom.

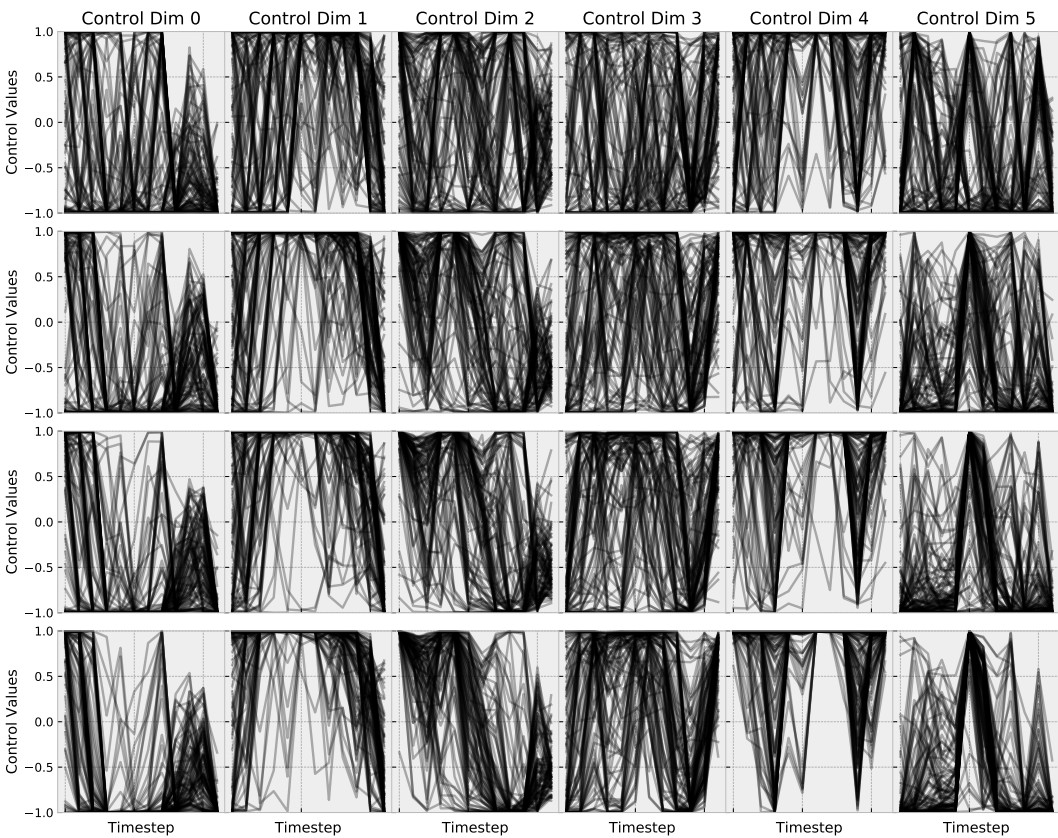

Figure 15: Visualization of the DCEM iterates on the walker to solve a single control problem starting from a random initial system state. The rows show iterates 1, 5, 7, 10 from the top to bottom.

