# OpenReview forum: "The Differentiable Cross-Entropy Method"
_ICLR.cc/2020/Conference — Reject_

### Official Review · AnonReviewer3 · 2019-10-21
**Official Blind Review #3**

**Rating:** 3

**Review:**

*Summary*
Various optimization methods can be wrapped to form black-box differentiable deep learning modules. This allows end-to-end learning of energy functions that can be used, for example, in continuous control. There is a whole cottage industry of designing these modules. Advancements in this field are of broad interest to the ICLR community. This paper proposes to unroll the cross entropy method, which is very different than the standard practice of unrolling gradient descent. Experiments on continuous control benchmarks demonstrate that this can be used to learn a latent space in which test-time optimization is performed. By doing this optimization in latent space, it can be performed much faster than in the raw high-dimensional space.

*Overall Assessment*
The paper is well written and the technical contribution is explained well. Both evolutionary search methods (e.g. CEM) and unrolled gradient-based optimizers are very popular in ML these days. This paper will be of interest to many readers, since it works at the interface between these.

I do not have much background in continuous control, model-based RL, etc. Therefore, it is hard for me to assess whether the experiments compare to the right baselines, etc. It appears to me that the experiments on cheetah and walker do not compare a particularly broad set of methods. They only compare within the design space of DCEM. Furthermore, the key result in these experiments is that the DCEM policy results in more efficient inner optimization (such that running the policy is faster). The overall messaging of the paper was not about reducing the costs of executing the policy but in improving performance. Such a result is not provided in these experiments.

I have worked extensively with unrolled optimizers and can speak to the correctness and usefulness of the paper's methodological contribution.and the experiments in sec 5.1. However, these are more for providing insight into the method, and are not large-scale experiments.

My evaluation is a weak reject, since the paper would be greatly improved by stronger empirical results for the large-scale continuous control benchmarks with comparison to a broader set of methods.

*Comments*

The empirical advantage of DCEM vs. unrolled GD is clear, but it's not clear to me what the intuition behind this is. You write "one potential advantage of DCEM is that the output is more likely to be near a local minimum of the energy surface so that, e.g., more test-time iterations can be used to refine the solution." Why would GD not want the output to be be near a local minimum. Also, why is DCEM not also sensitive to the number of steps? The variance of the CEM distribution gives a natural lengthscale similar to the step size in GD. You discuss this further at the end of sec 5.1 Are there simple experiments you could do that compare the robustness of GD (such as with random restarts or unrolled Langevin dynamics) vs. DCEM?

In the standard CEM, doing weighted MLE with 0-1 weights coming from top-k is useful because the set of examples for MLE is size k, which yields computational savings. However, if you can tolerate doing weighted MLE on all available samples, then there may be better ways to set the weights than using a softened version of top-k. See, for example, the example weights in 'Design by Adaptive Sampling' (arxiv.org/abs/1810.03714). Can you comment on the suitability of other weighting schemes besides top-k? Also, will your relaxed top-k perform sensibly when there are many ties in the observed f(x) values?

The principal critique of the paper is the positive DCEM results on cheetah + walker are mostly about runtime, instead of performance. Can you speak more to why you don't think it is providing better performance as well? Perhaps the latent space is useful, for example, for transfer learning + adaptation?


**Experience Assessment:**

I have published in this field for several years.

**Review Assessment: Checking Correctness Of Derivations And Theory:**

N/A

**Review Assessment: Checking Correctness Of Experiments:**

I assessed the sensibility of the experiments.

**Review Assessment: Thoroughness In Paper Reading:**

I read the paper thoroughly.

---

> ### Author Response · Authors · 2019-11-13
> **Specific Response to R3 Part 1/1**
>
> Thanks very much for reading through our paper and giving us your thoughts on it. We are in agreement that our paper will be of interest to many readers, even in the current form. We have included a shared response in a separate thread here and below are some more specific responses to your review:
>
> > [Experiments] and "They only compare within the design space of DCEM."
>
> We have a more complete response to this in our shared response and would also like to clarify that our experiments are not just within the design space of DCEM. Our baselines on the cheetah and walker tasks match the SOTA performance of other control/RL architectures in this space.
>
> > The empirical advantage of DCEM vs. unrolled GD is clear, but it's not
> clear to me what the intuition behind this is.
>
> There are quite a few other interesting properties that we are investigating in followup work. For example, it seems useful to consider a distribution over large parts of the space that is being optimized over that is continually refined so that DCEM can consider a much broader range of possible solutions than gradient descent does, and it isn’t as susceptible to immediately focusing on local solutions to the problem.
>
> Another direction of followup work we are pursuing is that the comparison of DCEM vs GD does not have to be so binary and one could consider unrolled optimizers that use both zeroth- and first-order information and thus could capture either DCEM or GD as special cases.
>
> > Why would GD not want the output to be near a local minimum?
>
> There may be too many degrees of freedom, especially in the energy-based setting when random restarts or other additional modifications are not used. As our Figure 1 shows, one case is when the GD iterates always start at a high-energy location and the energy surface learns to make the iterate descend to the regression target.
>
> > Also, why is DCEM not also sensitive to the number of steps?
>
> DCEM can still be sensitive to the number of steps -- in the synthetic task we’re considering, it may not be as sensitive to the number of steps and results in a local optimum around the data because we initialize the sampling distributions to cover the entire range of the output space. Thus the energy function does not have as many degrees of freedom as with vanilla GD, which one cannot start with a distribution over the entire output space that’s refined around the optimum.
>
> > Alternatives to top-k / “Design by Adaptive Sampling”  (https://arxiv.org/abs/1810.03714)
>
> Thanks for the reference! We were not aware of that paper and have added a reference to it. Upon a quick read-through it is not immediately clear to us how to apply it in our setting. In their section (2) they discuss how they relax the likelihood problem in their eq (12) with eq (13) by relaxing the set S to S^(t) by using some stochastic oracle to estimate p(y \geq \gamma^(t) | x). It’s not immediately obvious to us how to create an oracle like this in our setting, as in their Appendix S5, they are learning neural networks for the oracles in their setting. Would we also have to learn an oracle in our setting too? If so, one nice property of using the soft top-k relaxation we are using is that it does not involve any additional learning.
>
> > Can you comment on the suitability of other weighting schemes besides top-k?
>
> In most cases we are interested in using CEM, the top-k operation is used as the weighting scheme and we are not very familiar with any reasonable alternatives that are used in the optimization setting we consider here.

---

> > ### Author Response · Authors · 2019-11-13
> > **Specific Response to R3 Part 2/2**
> >
> > > Also, will your relaxed top-k perform sensibly when there are many ties in the observed f(x) values?
> >
> > Yes, the soft top-k operations can nicely handle ties like this and will do something reasonable. To give a quick code example here using the LML code from https://github.com/locuslab/lml, we can look at some example inputs and outputs here for the soft top-3 operation:
> >
> > ```
> > import lml
> > import torch
> >
> > for i in range(4):
> >     x = torch.ones(5)
> >     if i > 0 :
> >         x[-i:] = -1.
> >     y = lml.LML(3)(x)
> >     print(f'x = {x.numpy()}\ny = {y.numpy()}\n==========')
> >
> > x = [1. 1. 1. 1. 1.]
> > y = [0.599997 0.599997 0.599997 0.599997 0.599997]
> > ==========
> > x = [ 1.  1.  1.  1. -1.]
> > y = [0.6917577  0.6917577  0.6917577  0.6917577  0.23296393]
> > ==========
> > x = [ 1.  1.  1. -1. -1.]
> > y = [0.78206927 0.78206927 0.78206927 0.3269012  0.3269012 ]
> > ==========
> > x = [ 1.  1. -1. -1. -1.]
> > y = [0.8497297  0.8497297  0.43351656 0.43351656 0.43351656]
> > ```
> >
> > One numerical edge case of vanilla CEM is when all of the top-k values are the same, the variance becomes near-zero and no further updates are necessary. If this happens with DCEM we just return the current iterate and hope that it is either optimal for the task at hand, or that the parameters of the objective can be updated to continue doing parameter learning. In practice, we have not noticed our checks/warnings being thrown for this edge case in any of our experiments.
> >
> > > Can you speak more to why you don't think it is providing better performance as well?
> >
> > The baselines for the task we are considering are already near-SOTA and there are a number of additional ablations that we plan on doing that we’ve put in the shared response portion. We also needed to significantly reduce the number of trajectories the controller can sample to differentiate through it.
> >
> > > Perhaps the latent space is useful, for example, for transfer learning + adaptation?
> >
> > Yes, the latent control sequence space is likely to pick up on some shared task-agnostic structures although it may not be directly useful for transfer. For example, the structure of the optimal control sequence space for making the humanoid run is likely very different that the structure of the optimal control sequence space for making the humanoid jump or perform other tasks, although the two spaces may contain some similarities such as smoothness over time or correlations behind how the actuators can work together.

---

### Official Review · AnonReviewer2 · 2019-10-23
**Official Blind Review #2**

**Rating:** 3

**Review:**

After reading authors' response, I am sticking to my original decision. Authors addressed most of the issues I raised and I am happy with their response; however, I still believe the paper should not be accepted since it is not adding enough value. The problem is important and impactful. However, the algorithmic idea comes from LML (Amos 2019), and the impact on the real problems has not been demonstrated. Hence, it is adding no value algorithmically, and adding a very small value from application perspective. It is basically saying LML can be trivially applied to differentiate through CEM, and it works on some simple toy problems. To me this is mostly a sanity check. Hence, I am sticking to my weak-reject decision.
-------
The manuscript is proposing a method to make cross-entropy method (CEM) differentiable. CEM is a widely used zeroth-order optimization method. The main idea in the paper is applying the recently proposed limited multi-label projection (LML) layer in a straight-forward manner to the CEM since the major computational tool in CEM iteration is top-k selection. The authors apply the proposed method to synthetic energy-based learning and continuous control problems.

The proposed method is definitely impactful. Considering the fact that CEM is a powerful and widely used tool, I believe the work will lead to many interesting follow-ups. In addition to these, the work is addressing computational scalability of model-based RL which is both under-explored and important problem.

The proposed model is novel from a modelling perspective since it makes CEM part of end-to-end learnable models. Whereas, it has no algorithmic novelty since it is a straightforward application of the LML layer to the CEM problem. Lack of algorithmic novelty is not an issue but the authors should at least discuss similarities to LML (Amos 2019) in a clear manner in related work. Not including it in the related work is somewhat surprising to me.

The exposition can clearly be improved. First of all, Proposition 1 is an existing result, hence authors should give a proper citation in its definition. Second of all, Proposition 3 does not include anything about asymptotic (tau -> 0) whereas the stated one-line proof is using asymptotic arguments. Finally, there are other minor issues like Lemma1 not having a proof, proposition 1 has no statement about its proof etc.  The manuscript would significantly benefit from a thorough proof reading for mathematical completeness and correctness.

One major issue with the manuscript is the experimental study. 1) The only additional algorithmic element introduced by the manuscript is the tau and it is not experimented. Is it crucial to use the temperature parameter? If yes, what is the effect of it? Manuscript needs a collection of ablation studies discussing the tau. 2) The main claim of the paper is "...make solving the control optimization process significantly less computationally and memory expensive." This might be true but not really experimented. Authors do not report any quantitative computation time and/or memory requirement study. I believe the latent DCEM is more memory and computation efficient but quantifying this is important.

I am curious on the choice of CEM. There are other methods which can be utilized since this is basically a bi-level optimization problem. One can use implicit gradients or similar methods (like: https://arxiv.org/abs/1602.02355, https://arxiv.org/abs/1809.01465, https://arxiv.org/abs/1909.04630, http://proceedings.mlr.press/v22/domke12/domke12.pdf).  Can these methods also be utilized instead of back-propagation through optimization procedure? If yes, you should compare with them or explain why you did not. If no, you should explain why.

In summary, the paper is very impactful. On the other hand, the proposed empirical study significantly lacks in many aspects. I would be happy to increase my score if authors can address these issues.

**Experience Assessment:**

I have published one or two papers in this area.

**Review Assessment: Checking Correctness Of Derivations And Theory:**

I carefully checked the derivations and theory.

**Review Assessment: Checking Correctness Of Experiments:**

I assessed the sensibility of the experiments.

**Review Assessment: Thoroughness In Paper Reading:**

I read the paper thoroughly.

---

> ### Author Response · Authors · 2019-11-13
> **Specific Response to R2**
>
> Thanks for the encouraging review! We agree this paper can lead to many interesting followups have written some more thoughts on this in our shared response above. Here are some more specific responses to your review:
>
> > the authors should at least discuss similarities to LML (Amos 2019) in a clear manner in related work. Not including it in the related work is somewhat surprising to me.
>
> We considered adding the csoftmax/csparsemax/LML paper to the related work and are still open to having a discussion of them in there, but we see these methods as a tool for making the top-k operation differentiable here rather than an area of literature that we are building on.
>
> > First of all, Proposition 1 is an existing result, hence authors should give a proper citation in its definition. Second of all, Proposition 3 does not include anything about asymptotic (tau -> 0) whereas the stated one-line proof is using asymptotic arguments.
>
> Thanks, we have added some standard references for these well-known results and have clarified that the asymptotic is a corollary to Prop 3.
>
> > Finally, there are other minor issues like Lemma1 not having a proof, proposition 1 has no statement about its proof etc.  The manuscript would significantly benefit from a thorough proof reading for mathematical completeness and correctness.
>
> We will add a proof of Lemma 1 using a standard epsilon-delta argument although we do not feel that this will significantly add to the paper as it is a trivial lemma with (as far as we can see) an uninsightful and generic proof. Do you suspect that there are any other mathematical completeness or correctness issues with our paper?
>
> We very strongly feel that the proof of this trivial lemma is not important to our paper, but if you feel otherwise and if your review of the paper would increase if we added this proof, please let us know and we will add it immediately rather than waiting until later in the review period.
>
> > 1) The only additional algorithmic element introduced by the manuscript is the tau and it is not experimented. Is it crucial to use the temperature parameter? If yes, what is the effect of it? Manuscript needs a collection of ablation studies discussing the tau.
>
> We ablated \tau for the cartpole experiments in our original submission and the results are in Appendix D. We do not see \tau as a crucial parameter and just added it to show that DCEM can become non-differentiable and can approach vanilla CEM as the temperature goes to zero. We use \tau=1 in all of our other experiments and it is not a very important hyper-parameter to tune as the learning is likely able to adapt to any reasonable value of it.
>
> > 2) The main claim of the paper is "...make solving the control optimization process significantly less computationally and memory expensive." This might be true but not really experimented. Authors do not report any quantitative computation time and/or memory requirement study. I believe the latent DCEM is more memory and computation efficient but quantifying this is important.
>
> One main claim of our paper is that we can create a differentiable controller with CEM. This is impossible to do if DCEM is applied to the original control problem as it usually is. We do quantify this in the paper, as, for example, using CEM with 1000 samples in each iteration for 10 iterations with a horizon length of 12 requires 120,000 evaluations of the transition dynamics to obtain the next action for a *single* state. Trying to keep track of these evaluations and backprop through all of these causes OOM issues. We are able to reduce this by an order of magnitude to “just” 12,000 transition dynamics evaluations which enables us to differentiate through them.
>
> > CEM vs implicit differentiation
>
> This is definitely interesting and relevant work to ours, some of which we cited and discussed in the original version of our submission. We have updated our submission to include all of these references. The crux of the issue in the control setting is that reaching a fixed point to implicitly differentiate through can be extremely difficult, especially for the non-linear dynamical systems that we consider in this paper with approximate neural network dynamics. The differentiable MPC paper (http://papers.nips.cc/paper/8050-differentiable-mpc-for-end-to-end-planning-and-control) implicitly differentiates through a non-convex continuous control problem by reaching a fixed-point in SQP iterates and then differentiating through that locally convex approximation to the control problem. However they only considered simple and smooth settings where reaching a fixed point almost always happened and their method does not work if a fixed point is not reached, and thus we are unable to compare to them. In contrast our method works even when a fixed point is not reached.

---

### Official Review · AnonReviewer1 · 2019-10-23
**Official Blind Review #1**

**Rating:** 3

**Review:**

This paper proposes a differentiable variant of the Cross-Entropy method and shows its use for a continuous control task.
- It introduces 4 hyper-parameters and it is not clear how robust the method is to these.
- Although the idea is interesting, I think the paper needs a more rigorous experimental comparison with previous work and other methods.
Detailed review below:
- The abstract should mention clearly that the proposed method allows you to differentiate through argmin operation and can be used for end to end learning. Similarly, please reframe parts of the introduction to make it more accessible to a general reader. For example, in the introduction,  "approximation adds significant definition and structure to an otherwise...". This statement requires more context to make it useful. Similarly, "smooth top-k operation" is not clear.
- Is there a way to guarantee that the solution found by (D)CEM is a reasonable approximation to the argmin. For unrolled gradient descent, this can be done by looking at the gradient wrt x.
- It might be more useful to explain CEM before the related work section or just moving the related work to the end.
- Section 3: If the paper is about CEM, please give some motivation and details rather than just citing De Boer, 2005.
- There is a notation clash between \pi for the sort and policy later in the paper. Similarly, "t" is for both for the iterations of CEM and the time-stamp in the control problem.
- I don't understand how Proposition 1 adds to the paper. This is a standard thing. Similarly for Proposition 3.
- Isn't there an easier way to make the top-k operation soft - by sampling without replacement proportional to the probabilities? Please justify this design decision. Similarly, how is the temperature \tau chosen in practice?
- Please explain the paragraph: "Equation 4 is a convex optimization layer and... GPU-amenable.." Isn't this critical to the overall scalability of this method?
- - How are the hyper-parameters for CEM  chosen - the function g(.), the value of k, \tau, T chosen in practice. If the criticism of GD is that it overfits to the hyper-parameters - learning rate and the number of steps, why isn't this a problem with (D)CEM.
- Section 4: Since you're comparing against unrolled GD, please formally state what the method is.
- Section 4.2: How is the structure of Z decided, that is how do you fix the space for searching for the policy in the Z space?
- There are other methods that auto-encode the policy u_1:H to search the space. How does the proposed method compare to these methods? This is important to disentangle the effect of GD vs CEM and that of just searching in a more tractable space of policies.
- Section 5.1: How is the number of optimizer steps (=10) decided? Also, how is the learning rate for GD picked. Is the performance of unrolled GD worse for all values of \eta, even after a grid-search over the learning rates?
- For Section 5.2, please compare to baselines mentioned in the paper. Also, there needs to be an ablation/robustness study for the DCEM method.






**Experience Assessment:**

I do not know much about this area.

**Review Assessment: Checking Correctness Of Derivations And Theory:**

I carefully checked the derivations and theory.

**Review Assessment: Checking Correctness Of Experiments:**

I carefully checked the experiments.

**Review Assessment: Thoroughness In Paper Reading:**

I read the paper thoroughly.

---

> ### Author Response · Authors · 2019-11-13
> **Specific Response to R1**
>
> Thank you for giving our paper a close read and for the detailed comments despite it being out of your area.  We have posted shared response in a separate thread, and here are some more specific responses to your review.
>
> > "I don't understand how Proposition 1 adds to the paper. This is a standard thing. Similarly for Proposition 3."
>
> We agree that these are trivial propositions. They are helpful to the paper, as the solution to Prop 3 may not be immediately obvious to all readers (and is not shown in exactly that form in other references) and we think the connection to Prop 1 is interesting. We have added citations around these to help clarify that we are not claiming to be the original source of these well-known facts.
>
> > Is there a way to guarantee that the solution found by (D)CEM is a reasonable approximation to the argmin. For unrolled gradient descent, this can be done by looking at the gradient wrt x.
>
> This is an interesting point, and not something that people usually check even when unrolling gradient descent. With CEM and DCEM, one could check and see if all of the iterates are the same value.
>
> > Similarly, how is the temperature \tau chosen in practice?
>
> We introduced the \tau hyper-parameter just to show that DCEM can approach the vanilla CEM as \tau approaches 0. In all of our main experiments we use \tau=1 and do not think that this is a very important hyper-parameter empirically as the function being learned should be able to adapt to whatever is being learned, as long as it starts reasonably far away from the hard top-k operation.
>
> > How are the hyper-parameters for CEM chosen - the function g(.), the value of k, \tau, T chosen in practice. If the criticism of GD is that it overfits to the hyper-parameters - learning rate and the number of steps, why isn't this a problem with (D)CEM.
>
> There’s a lot of room for choosing hyper-parameters here and selecting hyper-parameters is the bane of a lot of research and there is a lot to discuss in this space. We will keep our response here short as our rebuttal is already quite long, but we note that in many domains, such as for control, these hyper-parameters still have to be selected for vanilla CEM and a good starting point for our differentiable variant in these domains is to use similar values.
>
> > Section 4: Since you're comparing against unrolled GD, please formally state what the method is.
>
> Thanks, we have formalized this in our paper at the beginning of the section.
>
> > Section 4.2: How is the structure of Z decided, that is how do you fix the space for searching for the policy in the Z space?
>
> We assume that Z is some low-dimensional Euclidean space/box and we learn a decoder that maps these points back up to the full control sequence space.
>
> > There are other methods that auto-encode the policy u_1:H to search the space. How does the proposed method compare to these methods? This is important to disentangle the effect of GD vs CEM and that of just searching in a more tractable space of policies.
>
> Yes, we included references to Co-Reyes et al. (2018); Antonova et al. (2019) in our related work section and please let us know if there are any others you know of. Our work is complementary to these methods and can be used on top of them to help fine-tune their learned latent space if you have the knowledge that their latent space is going to be used for control.
>
> > Section 5.1: How is the number of optimizer steps (=10) decided? Also, how is the learning rate for GD picked. Is the performance of unrolled GD worse for all values of \eta, even after a grid-search over the learning rates?
>
> In all of our experiments for our paper we use unroll 10 steps of GD or DCEM as it is a relatively standard number to use in these settings, and we arbitrarily set the GD learning rate to something that is also relatively standard here. Our goal in this setting is not to show that DCEM can over-fit to the small synthetic regression task we are considering and give superior performance to GD -- in fact the performances between GD and DCEM are nearly identical -- and instead our goal is to show a setting where GD and DCEM perform the same but learn extremely different energy surfaces, which we show in Figure 1. Do you agree that this is a novel demonstration of this happening? If not, can you send us over references with similar ideas so that we can properly contextualize our work?
>
> > [Baselines/ablations]
>
> See our shared response above on differentiable control and SOTA here -- there are no easily applicable differentiable control baselines in the settings we consider as none of them work and we full-heartedly agree that more ablations/robustness studies of DCEM in this setting are important to study in future work as we use it as a more general policy class across a significantly wider range of environments, although we feel at this point for the purpose of the demonstration shown in this paper such ablations are not as insightful

---

### Public Comment · ~Zhiao_Huang1 · 2019-09-30
**Great work; Can we replace soft top-k with any other soft-attetion mechanism?**

I like this work very much. The differentiable optimization process is an important direction.

I wonder that if how important the soft top-k modular is here. The output of the soft-topk module is the weights over samples. The only requirement is that the sampler with a higher score should get higher weight. Can we calculate such weights with more straightforward methods? For example, when k=1, we can get such weights by softmax(v_t/\tau). It seems that such weights don't hurt the differentiability of the optimization process. I don't know if I am correct. I will appreciate if the authors could give some words on this.

---

> ### Author Response · Authors · 2019-09-30
> **On the weight calculation**
>
> Thank you for your comment. Yes, in theory, any function that computes a differentiable weighting from the sampled function values could be used as the weights in the maximum weighted likelihood problem (5) and most reasonable choices, such as the softmax, would likely work well in practice too. The end-to-end learning would likely be able to adapt to any reasonable differentiable weighting mechanism.
>
> We chose the LML projection because it captures the original cross-entropy method as a special case as the temperature approaches zero, which does not hold for a temperature-scaled softmax. The LML projection can also be done with a single line of code that, for the problems we consider in this paper (a batch of 128 problems with 100 variables and the soft top-10 entries) runs in ~6ms in comparison to the ~0.6ms the softmax takes.
>
> Something else to consider for future work -- one could imagine training a model using DCEM with a soft top-k operation and 1) slowly annealing the temperature to zero during training to help squeeze the last few bits of accuracy out of the system, or 2)  at evaluation time, using vanilla CEM with a hard top-k operation to get a slightly better solution to the optimization problem. In these cases, it may be more likely to help/work if a soft top-k operation like the LML projection is used rather than the softmax, for example, as it would be closer to the hard version.

---

> > ### Public Comment · ~Zhiao_Huang1 · 2019-10-01
> > **Thank you for the reply.**
> >
> > Yeah, soft top-k makes it be real "CEM" instead of something else. I guess top-k encourages exploration comparing with softmax. If hard version CEM is better than the soft version in general, LML should be a better choice here.

---

### Author Response · Authors · 2019-11-13
**Shared Response Part 1/2**

Thanks very much for the useful feedback and comments on our paper. We are especially happy to hear from R2 that "the proposed method is definitely impactful. Considering the fact that CEM is a powerful and widely used tool, I believe the work will lead to many interesting follow-ups" and from R3 that "this paper will be of interest to many readers, since it works at the interface between [evolutionary search] and [unrolled gradient descent]."

We have posted a new version of the paper that addresses some points raised by the reviewers (more specific details below) and emphasize that the goal of our paper is to present this idea with demonstrations across a variety of domains. We very much agree with the reviewers that there are many future directions to build on this work and that there is a near-unbounded space of possible future experiments, ablations, and analyses to be done, especially in the control and reinforcement learning setting. The ~10 pages of additional details and ablations we have in our appendix merely scratch the surface with what we believe to be among the most important and insightful pieces to include and are releasing all of our model, experiment, and plotting code so that anybody can quickly reproduce, extend, and analyze/ablate our approach in new ways that we haven’t considered.

# Differentiable control and SOTA results
We would firstly like to directly address the issue raised by all of the reviewers of our empirical results not being SOTA and not having comparisons to related approaches. The context of our contribution in the differentiable control literature is important to see the empirical value of our paper and is something that none of the reviewers commented on, and may have overlooked or under-appreciated. We would like to highlight and emphasize that *all* prior literature on differentiable control (e.g. https://arxiv.org/abs/1703.09260 https://arxiv.org/abs/1706.09597 https://arxiv.org/abs/1802.05803 https://arxiv.org/abs/1810.13400 https://openreview.net/forum?id=ryxC6kSYPr) have only shown empirical results in simple environments such as the pendulum and cartpole, may not work with neural network dynamics, and may not do policy learning.

Our empirical goal in this paper in this space is to demonstrate that DCEM can use policy learning to tune parts of a non-convex controller in more complex environments. We focused on this in the non-trivial DeepMind control suite cheetah and walker environments. The reviewers stated that our work is difficult to justify and compare to related work -- we agree this part could be made clearer in our paper, but our results are extremely consistent with previously published work in this space. Our baseline (which we call full CEM) is our implementation of PlaNet (https://arxiv.org/abs/1811.04551) in the proprioceptive setting and our baseline agents in the cheetah and walker proprioceptive environments are nearly identical to the results in the PlaNet paper. Our results are also extremely consistent with the agents published in the DMC paper https://arxiv.org/abs/1801.00690 and we have also evaluated them with 100 evaluation episodes. To further appreciate the difficulty of doing model-based control in these environments, you can see our model predictions and how policy learning with DCEM helps bring the agents back to more reasonable policies at: https://sites.google.com/view/diff-cross-entropy-method/home

In light of this new information and context, do the reviewers agree that this is a non-trivial novel demonstration that has not been shown in the prior differentiable control literature before? If not, can you please indicate why not and provide a reference to relevant differentiable control literature that makes a similar demonstration?

While we have been continuing to study and build upon CEM as a way of learning extremely generic controllers/policies that uniformly work across all of the standard continuous control environments, it is not the main focus of this paper, which we instead would like to be a thought-provoking new differentiable non-convex optimizer with broader applications. We are preparing many additional experiments and ablations in the control/RL space that would add at least 10-20 additional pages of details -- there are many interesting followup questions we are exploring. For example:

---

> ### Author Response · Authors · 2019-11-13
> **Shared Response Part 2/2**
>
> 1. DCEM can be used for model-free policies to help (semi-)amortize the max-Q computation in ideas such as https://openreview.net/forum?id=BkxXe0Etwr or model-based policies as shown in our paper.
> 2. How can DCEM be used with controllers that have a Q or value estimate function at the end such as https://arxiv.org/abs/1908.06012 This context is especially interesting as the controller's horizon approaches 0, this captures vanilla Q learning as a special case.
> 3. What is the best policy optimizer to use with a differentiable controller and what are the implications? While we show PPO experiments in this paper, one could also use SAC/TD3. However these algorithms are usually used in situations where the model-free policy and Q function have similar representational power (e.g. are neural networks) and it's not as clear if using a control-based policy with a neural network Q function is ideal, or if we can also use parts of the controller to essentially estimate a multi-step Q function.
> 4. Exploration is a large part of policy learning and we may be able to use the dynamics model to help with this, potentially by using disagreement or out-of-distribution detection as in https://arxiv.org/abs/1810.12162 and https://arxiv.org/abs/1906.04161
> 5. How should the controller be warm-started or given context? For a given system state, it's reasonable that the controller could immediately start with a guess of the region of the action sequence space that it could consider rather than starting with no information as we have done is this paper.
>
> We feel that including all of the details and experiments behind these directions in this version of the DCEM paper muddle the more general contribution here and are worth presenting separately. With this in mind, do the reviewers agree with our choice of showing demonstrations across a broader range of tasks in this version of the paper with careful comparisons in the RL/control setting in followup work?

---

### Decision · Program_Chairs · 2019-12-19

**Decision:**

Reject

**Comment:**

This paper proposes a differentiable version of CEM, allowing CEM to be used as an operator within end-to-end training settings. The reviewers all like the idea -- it is simple and should be of interest to the community. Unfortunately, the reviewers also are in consensus that the experiments are not sufficiently convincing. We encourage the authors to expand the empirical analysis, based on the reviewer's specific comments, and resubmit the paper to a future venue.